# Beyond Scalars: Evaluating and Understanding LLM Reasoning via Geometric Progress and Stability

**Xinyan Jiang** [1 2 3 4]  **Ninghao Liu** [5]  **Di Wang** [2]  **Lijie Hu** [1]

## Abstract

Evaluating LLM reliability via scalar probabilities often fails to capture the structural dynamics of reasoning. We introduce **TRACED**, a framework that assesses reasoning quality through theoretically grounded geometric kinematics. By decomposing reasoning traces into Progress (displacement) and Stability (curvature), we reveal a distinct topological divergence: correct reasoning manifests as high-progress, stable trajectories, whereas hallucinations are characterized by low-progress, unstable patterns (stalled displacement with high curvature fluctuations). Leveraging these signatures, our probabilistic framework achieves competitive performance and superior robustness across diverse benchmarks. Crucially, TRACED bridges geometry and cognition by mapping high curvature to "Hesitation Loops" and displacement to "Certainty Accumulation", offering a physical lens to decode the internal dynamics of machine thought.

## 1. Introduction

Large Language Models (LLMs) have demonstrated remarkable capabilities in complex reasoning, particularly through the generation of multi-step Chain-of-Thought (CoT) (Guo et al., 2025; Team, 2025; Abdin et al., 2025; Yang et al., 2025a). However, despite these advances, the reasoning process exhibits significant instability; models frequently suffer from hallucinations and logical fallacies, generating plausible-sounding but fundamentally incorrect derivations (Turpin et al., 2023; Shojaee et al., 2025; Huang et al., 2025).

Consequently, the ability to accurately assess the quality of a reasoning process, distinguishing valid deductions from confident fabrications, has become a critical challenge for reliable model deployment (Wu et al., 2024; Nguyen et al., 2024; Chen et al., 2025b; Yang et al., 2025c; Hu et al., 2025).

Existing reasoning evaluations bifurcate into two paradigms: External Assessment relying on supervision (Li et al., 2022; He et al., 2025; Li et al., 2023) and Internal Assessment utilizing intrinsic statistics (Xiong et al., 2023; Marjanović et al., 2025; Wang et al., 2024; Yang et al., 2026; 2025b). External Assessment typically employs auxiliary verifiers or annotations (Zhang et al., 2025c; 2024; Gandhi et al., 2025). While effective, their dependence on parametric supervised training or expert models precludes scalability during real-time inference (Sky et al., 2024). Conversely, Internal Assessment leverages label-free signals like probability or semantic entropy (Li et al., 2024; Farquhar et al., 2024). However, by reducing reasoning trajectories to static scalars, these methods discard critical temporal dynamics. Relying on point-wise aggregation (e.g., last-token probability) neglects the sequential evolution of thought, thereby missing structural signals essential for robust evaluation. Ultimately, both paradigms neglect the underlying reasoning mechanisms (Zhao et al., 2025; Bi et al., 2025), limiting their generalization and ability to distinguish justified certainty from hallucination. This leaves a gap for a framework that offers not just prediction but also a robust and transferable diagnosis.

To overcome the limitations of simple statistical metrics and provide stronger interpretability, recent research has turned to the geometry of hidden states to understand model behavior (Yao et al., 2025; Jiang et al., 2026; Kazama et al., 2026; Wang et al., 2026; Yu et al., 2026). Vilas et al. (2025) demonstrated that the temporal signals of the reasoning process contain rich information that can predict reasoning correctness. Complementing this, Zhou et al. (2025) theoretically established that reasoning behaves as a "geometric flow" controlled by logical structure, while Manson (2025) revealed that semantic concerns induce measurable curvature in metric-aligned spaces. Collectively, these works demonstrate that the geometric trajectory of hidden states is a structured manifestation of the reasoning process. How-

---

[1]Mohamed bin Zayed University of Artificial Intelligence (MBZUAI) [2]King Abdullah University of Science and Technology [3]Shanghai Advanced Research Institute, Chinese Academy of Sciences, Shanghai, China [4]University of Chinese Academy of Sciences, Beijing, China [5]Department of Computing, Hong Kong Polytechnic University, Hong Kong, China. Correspondence to: Lijie Hu <lijie.hu@mbzuai.ac.ae>.

*Proceedings of the 43rd International Conference on Machine Learning*, Seoul, South Korea. PMLR 306, 2026. Copyright 2026 by the author(s).

ever, current reasoning quality assessment methods fail to integrate these profound geometric insights; bridging these intrinsic geometric features with the practical challenge of reasoning quality assessment is of significant scientific value.

In this work, we introduce **T**rajectory **R**easoning **A**ssessment via **C**urvature **E**volution and **D**isplacement Dynamics (**TRACED**), a framework that assesses LLM reasoning quality through a geometric kinematics perspective. Specifically, to address the limitation of internal methods that reduce complex thought to simple scalars, we analyze the geometric properties of reasoning traces by decomposing them into **Progress** and **Stability**. We define *Progress* as the **displacement change** of the reasoning trajectory (where displacement change indicates significant thought progress) and *Stability* as the trajectory **curvature change** (where lower curvature change indicates higher thinking stability), as shown in Figure 1. These geometric features reveal a distinct geometric divergence: correct reasoning manifests as high progress and high stability (i.e., high magnitude and low curvature change), whereas incorrect reasoning is characterized by low progress and low stability (i.e., low displacement and high curvature change). By deriving a label-calibrated contrastive subspace, this topological divergence establishes a natural distributional separation, enabling us to distinguish reasoning quality purely through latent dynamics. To overcome the computational burden and poor generalization of external assessment, we leverage these features to construct a Bayesian probabilistic model. This model performs direct, latent dynamics evaluation of reasoning quality by exploiting the distributional separation between correct and incorrect reasoning in geometric space. Furthermore, to bridge the gap between geometry and cognitive thinking, we map these geometric features to cognitive states (e.g., Reflection, Exploration, Certainty). We mechanistically interpret high curvature change as the physical manifestation of a "Hesitation Loop" ( an oscillation between exploration and reflection), while high displacement change reflects the accumulation of certainty as concept transitions converge toward the final answer.

Comprehensive evaluations have been conducted across four models (including Instruction-tuned LLMs (Team et al., 2024; Grattafiori et al., 2024) and Large Reasoning Models (LRMs) (Guo et al., 2025; Yang et al., 2025a)) to validate the effectiveness of TRACED. We employ AUROC (Boyd et al., 2013), AUPR, and FPR@95 (Manning & Schütze, 1999) to evaluate the effectiveness of geometric features in distinguishing between correct and incorrect reasoning paths. Our experiments span six benchmarks in two domains: (1) **Structured Reasoning**: GSM8K (Cobbe et al., 2021), MATH (Hendrycks et al., 2021), TheoremQA (Chen et al., 2023), GPQA (Rein et al., 2024); and (2) **Open-Ended Reasoning**: Social IQA (Sap et al., 2019), Understanding Fables

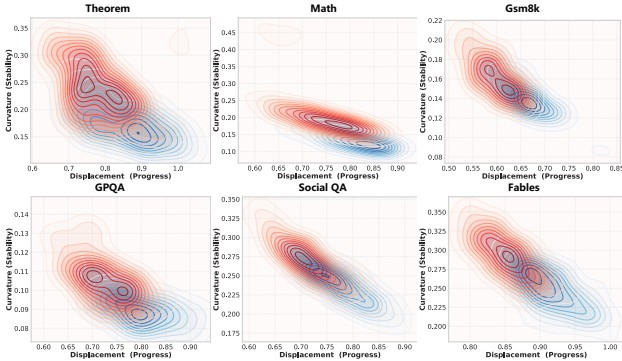

*Figure 1.* **Topological Divergence of Reasoning Quality.** Joint distribution of net displacement ($M$) and curvature ($K$) across Structured and Open-Ended domains. The visualization confirms a consistent separation: correct reasoning traces (blue) exhibit a high-displacement, low-curvature pattern, while incorrect chains (red) are characterized by low-displacement stagnation and high-curvature oscillations.

(Srivastava et al., 2023). Our framework demonstrates superior performance across diverse benchmarks, confirming geometric features as a reliable and robust indicator of reasoning quality.

Our contributions can be summarized as follows:

- **Geometric Decomposition:** We evaluate the quality of reasoning by leveraging theoretically grounded geometric signatures (Displacement and Curvature), establishing that valid reasoning is characterized by high-progress, stable trajectories, whereas hallucinations exhibit low-progress, unstable geometric patterns.

- **Latent Kinematics Assessment:** Constructs a probabilistic model leveraging geometric kinematics within a label-calibrated subspace, achieving competitive performance and superior robustness across benchmarks.

- **Geometric-Cognitive Correspondence:** We bridge geometry and cognition by mapping geometric features to hidden states, interpreting high curvature as "Hesitation Loops" and high displacement as "Certainty Accumulation," thereby enhancing the interpretability of the reasoning process.

## 2. Preliminaries

### 2.1. Reasoning as a Trajectory in Latent Space

Formally, let a Large Language Model be denoted as a function $f : \mathcal{X} \to \mathcal{Y}$ parameterized by $\theta$. Given an input query $\mathbf{x}$, the model generates a reasoning chain (Chain-of-Thought) consisting of $T$ tokens, $\mathbf{y} = (y_1, y_2, \ldots, y_T)$, followed by a final answer.

We use the hidden state of the final layer (immediately preceding the unembedding head) at each time step $t$. Let $\mathbf{h}_t \in \mathbb{R}^d$ denote this latent representation for the $t$-th token

$y_t$. Consequently, the entire reasoning process is formalized as a discrete time-series trajectory $\mathcal{H}$ :

$$\mathcal{H} = \mathbf{h}_1 \xdashrightarrow{} \mathbf{h}_2 \xdashrightarrow{} \ldots \xdashrightarrow{} \mathbf{h}_T \qquad (1)$$

Here, the global geometry of $\mathcal{H}$ encodes the structural properties of the reasoning process.

---

**Algorithm 1** Get Reasoning Quality Space

---

1: **Input:** Sets of hidden state trajectories $\mathcal{D}_{\text{pos}}$ and $\mathcal{D}_{\text{neg}}$, where each sample $d_n \in \mathcal{D}$ is a sequence $\mathbf{d}_n = (\mathbf{h}_{n,1}, \ldots, \mathbf{h}_{n,T})$; Induced Metric $G = W_U^\top W_U$.

2: **Step 1: Semantic Whitening.** Compute isotropized states for each sample $n$ using the metric square root:
$\mathbf{h}'_{n,t} \leftarrow G^{1/2} \mathbf{h}_{n,t}$

3: **Step 2: Differential Dynamics.** Capture kinematic updates in the whitened space:
$\Delta \mathbf{h}'_{n,t} \leftarrow \mathbf{h}'_{n,t} - \mathbf{h}'_{n,t-1}$

4: **Step 3: Sample Covariance.** Compute kinematic variance for sample $n$:
$C_n \leftarrow \frac{1}{T-1} \sum_t \Delta \mathbf{h}'_{n,t} (\Delta \mathbf{h}'_{n,t})^\top$

5: **Step 4: Contrastive Aggregation.**
$C^+ \leftarrow \frac{1}{|\mathcal{D}_{\text{pos}}|} \sum_{\mathbf{d}_n \in \mathcal{D}_{\text{pos}}} C_n$
$C^- \leftarrow \frac{1}{|\mathcal{D}_{\text{neg}}|} \sum_{\mathbf{d}_n \in \mathcal{D}_{\text{neg}}} C_n$
$S \leftarrow C^+ - \lambda C^-$ (where $\lambda = \|C^+\|_F / \|C^-\|_F$)

6: **Step 5: Basis Extraction.**
Eigendecompose $S \rightarrow B$ (Top-$k$ eigenvectors)

7: **Output:** Reasoning Quality Space Basis $B$

---

### 2.2. The Execution Manifold and Semantic Geometry

To accurately capture the geometric dynamics of reasoning, we must define a space where geometric movement strictly corresponds to semantic evolution.

**The Problem: Geometry-Semantics Mismatch.** Directly measuring geometric features (e.g., displacement) in the raw hidden state space $\mathbb{R}^d$ is problematic due to the "anisotropy" of the representation. The raw space is often dominated by high-frequency outlier dimensions or systematic noise that does not contribute to the model's actual predictions. In this uncorrected space, a large Euclidean distance does not necessarily imply a significant change in meaning. If we compute dynamics directly on $\mathbf{h}_t$, our metrics (Displacement and Curvature) would likely measure numerical noise rather than the progress of thought (Timkey & van Schijndel, 2021).

To resolve this, we must measure the trajectory in the "vocabulary space", the only space where the model's internal states translate into actual meaning. Following Manson (2025), we adopt the semantic metric induced by the model's unembedding matrix $W_U$. we define the **induced metric tensor** $G = W_U^\top W_U$ to gauge the magnitude of state vectors. This allows us to measure geometric changes under the

norm $\|\mathbf{v}\|_G = \sqrt{\mathbf{v}^\top G \mathbf{v}} = \|W_U \mathbf{v}\|_2$, effectively weighting the hidden state dimensions by their impact on the vocabulary distribution. By utilizing this induced metric, we filter out non-semantic artifacts. This ensures that any measured geometrical changes reflect a genuine shift in the probability distribution over the vocabulary, providing a rigorous physical ground for our reasoning metrics.

## 3. Method: TRACED

TRACED is a framework that evaluates reasoning via geometric kinematics. It operates in three steps: (1) transforming states into a discriminative quality space to reduce noise (Sec.3.1); (2) measuring Displacement (progress) and Curvature (stability) as geometric signatures (Sec.3.2); and (3) employing a Bayesian model to diagnose reliability based on the topological divergence (Sec.3.3).

### 3.1. Constructing the Reasoning Quality Space

Building on the **semantic geometry** defined in Sec.2.2, to accurately measure reasoning quality, we must isolate the dynamics of logical deduction from unrelated factors (e.g., syntax or static facts) present in the hidden states.

We construct a reasoning quality discriminative space $B$ that maximizes the differences between correct and incorrect reasoning as shown in Algorithm 1. Let $\mathcal{D}_{\text{pos}}$ and $\mathcal{D}_{\text{neg}}$ denote sets of correct and incorrect reasoning chains. Conceptually, LLM hidden states entangle logical deduction with shared generative factors. Since valid ($C^+$) and invalid ($C^-$) trajectories share this fundamental linguistic base, raw variance fails to isolate true logic. Thus, we compute the scaled difference between their kinematic covariance matrices, $S = C^+ - \lambda C^-$, to explicitly cancel these shared non-logical directions and identify directions where effective reasoning evolves most distinctly. Moreover, the scaling factor $\lambda = \|C^+\|_F / \|C^-\|_F$ normalizes geometric energy. Since incorrect paths' "hesitation loops" yield vastly larger covariances than stable correct paths, unscaled subtraction would be dominated by hallucination magnitudes. This normalization strictly isolates structural and directional differences, not mere amplitude. We then extract the top-$k$ eigenvectors (fixing $k = 8$, with sensitivity analyzed in Appendix M) to form the basis $B$, capturing the principal dimensions that maximally distinguish correct from incorrect reasoning.

The final state $\mathbf{z}_{n,t}$ corresponding to the $t$-th token of a given sample $d_n \in \mathcal{D}$ is obtained by transforming the raw hidden state $\mathbf{h}_{n,t}$ via the metric-induced feature map $G^{1/2}$ and projecting it onto the quality subspace basis $B$:

$$\mathbf{z}_{n,t} = B^\top (G^{1/2}) \mathbf{h}_{n,t} \qquad (2)$$

All subsequent geometric metrics are computed using these

projected coordinates $\mathbf{z}_{n,t}$.

## 3.2. Geometric Signatures of Reasoning Quality

**Theoretical Motivation.** Some studies observe that correct reasoning is characterized by a structured, efficient evolution of semantic states, whereas hallucinations or logical errors often correlate with thinking behavior anomalies, such as local stagnation or erratic directional shifts (Sun et al., 2025; Bi et al., 2025). Building on these findings, we advance the understanding of reasoning dynamics from empirical observations to a framework of geometric characterization and theoretical rigorousness. Specifically, we quantify reasoning quality via these two distinct physical components:

**Progress:** Does the process generate deterministic information shifts, effectively accumulating certainty?

**Stability:** Is the logical flow stable, maintaining a consistent direction, or is it exhibiting volatile orientation changes?

To operationalize these physical concepts, we map them to specific geometric features motivated by our theoretical analysis. As formulated in Appendix E, under an idealized stochastic differential equation framework, *Displacement* and *Curvature* naturally emerge as effective geometric proxies for characterizing reasoning quality. Consequently, we define:

**1. Displacement (Progress).** We quantify the "Progress" of reasoning as the normalized net geometric distance traversed in the representation space. First, we define the local update vector at step $t$ as $\Delta\mathbf{z}_{n,t} = \mathbf{z}_{n,t} - \mathbf{z}_{n,t-1}$. We quantify progress using the Normalized Net Displacement $M_n$:

$$M_n = \frac{1}{T}\|\mathbf{z}_{n,T} - \mathbf{z}_{n,0}\|_2 = \frac{1}{T}\left\|\sum_{t=1}^{T}\Delta\mathbf{z}_{n,t}\right\|_2 \quad (3)$$

*Physical Interpretation:* This metric reflects the **Progress of Thought**. A high displacement ($M_n \gg 0$) implies that the model is confidently transitioning between distinct semantic states, effectively "accumulating certainty" towards a conclusion. Conversely, low displacement suggests the model is idling, repeating information, or stalling without substantive semantic progress.

**2. Curvature (Stability).** We measure reasoning "Stability" via geometric curvature. Following discrete differential geometry, we define velocity $\mathbf{v}_t = \Delta\mathbf{z}_{n,t}$ and acceleration $\mathbf{a}_t = \Delta\mathbf{z}_{n,t+1} - \Delta\mathbf{z}_{n,t}$. The extrinsic curvature $\kappa_{n,t}$ is computed as:

$$\kappa_{n,t} = \frac{\sqrt{\|\mathbf{v}_t\|_2^2\|\mathbf{a}_t\|_2^2 - (\mathbf{v}_t \cdot \mathbf{a}_t)^2}}{\|\mathbf{v}_t\|_2^3 + \epsilon} \quad (4)$$

where $\epsilon$ is a small constant term for numerical stability. The Average Trajectory Curvature $K_n$ is then averaged over interior points:

$$K_n = \frac{1}{T-2}\sum_{t=1}^{T-2}\kappa_{n,t} \quad (5)$$

*Physical Interpretation:* This metric reflects the **Stability of Reasoning**. High curvature ($\kappa \gg 0$) indicates sharp semantic turns or oscillations (instability), while low curvature implies a smooth deduction trajectory. Normalizing against $\|\mathbf{v}\|_2^3$ ensures scale-invariance and robustness against noise.

**Geometric Modes.** To empirically validate the effectiveness of these features, we visualized the joint distribution of normalized displacement ($M_n$) and curvature ($K_n$) across a diverse set of reasoning benchmarks on DeepSeek-R1-Llama-8B (experimental settings detailed in Appendix A). As shown in Figure 1, we observed a consistent topological separation between correct and incorrect reasoning traces across all domains; consistent results for other models are shown in Appendix B. This pattern is robust across architectures and reasoning types, provides a robust physical signature for distinguishing reasoning quality:

**Correct Reasoning:** High-quality chains consistently cluster in the **high-displacement, low-curvature** ($M \uparrow, K \downarrow$) regime. This geometric pattern indicates that the model is effectively accumulating information and progressing directly toward the solution without significant backtracking or hesitation.

**Incorrect Reasoning:** Conversely, reasoning errors and hallucinations cluster in the **low-displacement, high-curvature** ($M \downarrow, K \uparrow$) regime. This pattern indicates that the model engages in excessive reflection or repeats redundant steps, resulting in local stagnation and frequent changes in direction without making substantial semantic progress toward the answer.

### 3.3. Gaussian Discriminant Analysis for Assessment

Having established that correct and incorrect reasoning trajectories exhibit distinct topological signatures (separation in $M$-$K$ features), we can now formalize quality assessment as a probabilistic classification problem. We leverage the low-dimensional geometric features to perform Maximum A Probability (MAP) estimation.

**Probabilistic Formulation.** Let $\mathbf{x}_n = [M_n, K_n]^\top$ denote the geometric feature vector derived from the reasoning manifold for sample $n$. Let $y_n \in \{1, 0\}$ be the latent quality label (1 for Correct/High Quality, 0 for Incorrect/Low Quality). According to Bayes' theorem, the posterior probability of a trajectory being correct is:

$$P(y_n = 1|\mathbf{x}_n) = \frac{P(\mathbf{x}_n|y_n = 1)P(y_n = 1)}{P(\mathbf{x}_n)} \quad (6)$$

**Likelihood** $P(\mathbf{x}_n|y_n = c)$**:** We approximate the feature density as a Gaussian $\mathcal{N}(\mu_c, \Sigma_c)$. This choice is grounded in

*Table 1.* Comparison of reasoning quality assessment methods across multiple models and tasks. See Appendix N for statistical uncertainty analysis (95% CIs) confirming the stability of these results.

| | Fables | | | GPQA | | | GSM8K | | | MATH | | | Social_iqa | | | Theorem | | |
|---|---|---|---|---|---|---|---|---|---|---|---|---|---|---|---|---|---|---|
| Method | AUROC↑ | AUPR↑ | FPR@95↓ | AUROC↑ | AUPR↑ | FPR@95↓ | AUROC↑ | AUPR↑ | FPR@95↓ | AUROC↑ | AUPR↑ | FPR@95↓ | AUROC↑ | AUPR↑ | FPR@95↓ | AUROC↑ | AUPR↑ | FPR@95↓ |
| *DeepSeek-R1-Llama-8B* | | | | | | | | | | | | | | | | | | |
| MSP | 0.6044 | 0.6427 | 0.8919 | 0.3837 | 0.4750 | 0.8887 | 0.7424 | 0.7425 | 0.8235 | 0.6276 | 0.6203 | 0.8125 | 0.5867 | 0.6237 | 0.9194 | 0.5270 | 0.4951 | 0.8737 |
| Perplexity | 0.6719 | 0.6581 | 0.8568 | 0.3831 | 0.4107 | 0.8767 | 0.6096 | 0.6088 | 0.8824 | 0.5808 | 0.5628 | 0.8688 | 0.5911 | 0.6174 | 0.8065 | 0.5229 | 0.5552 | 0.8211 |
| Entropy | 0.6156 | 0.6481 | 0.8378 | 0.4857 | 0.5159 | 0.8823 | 0.6863 | 0.6916 | 0.8235 | 0.6041 | 0.5921 | 0.8875 | 0.5925 | 0.6095 | 0.8194 | 0.5111 | 0.4951 | 0.8737 |
| LR Probe | 0.7177 | 0.6539 | 0.8297 | 0.7588 | 0.5451 | 0.8571 | 0.7995 | 0.8195 | 0.7059 | 0.7471 | 0.6541 | 0.8625 | 0.7097 | 0.6883 | 0.7091 | 0.8435 | 0.6497 | 0.8158 |
| SAPLMA | 0.6944 | 0.6555 | 0.8919 | 0.7180 | 0.5505 | 0.7143 | 0.7996 | 0.8204 | 0.6824 | 0.7161 | 0.6363 | 0.8000 | 0.6957 | 0.6891 | 0.7097 | 0.8518 | 0.6617 | 0.8421 |
| CoE | 0.5856 | 0.5603 | 0.8649 | 0.5000 | 0.6000 | 0.8317 | 0.5651 | 0.5653 | 0.8824 | 0.6156 | 0.6332 | 0.8812 | 0.6465 | 0.6338 | 0.8226 | 0.6053 | 0.6402 | 0.8474 |
| CoT-Kinetics | 0.7162 | 0.5787 | 0.8627 | 0.5490 | 0.5000 | 0.8500 | 0.6194 | 0.5527 | 0.8326 | 0.6755 | 0.6271 | 0.7933 | 0.6738 | 0.6010 | 0.8488 | 0.6738 | 0.5951 | 0.8133 |
| TRACED | 0.7191 | 0.6586 | 0.8242 | 0.8300 | 0.6607 | 0.6400 | 0.8061 | 0.8283 | 0.6500 | 0.7489 | 0.6549 | 0.7500 | 0.7536 | 0.6909 | 0.6500 | 0.8730 | 0.7094 | 0.7625 |
| *Qwen3-4B-Thinking-2507* | | | | | | | | | | | | | | | | | | |
| MSP | 0.6509 | 0.6727 | 0.8462 | 0.6000 | 0.6787 | 0.8000 | 0.6509 | 0.6489 | 0.8654 | 0.4844 | 0.5592 | 0.7500 | 0.6741 | 0.6441 | 0.8036 | 0.3273 | 0.4622 | 0.8091 |
| Perplexity | 0.5503 | 0.6344 | 0.8675 | 0.5600 | 0.5450 | 0.6000 | 0.6191 | 0.5917 | 0.8077 | 0.6094 | 0.5602 | 0.7650 | 0.6159 | 0.6048 | 0.8750 | 0.6273 | 0.5554 | 0.6364 |
| Entropy | 0.6450 | 0.6549 | 0.8462 | 0.5600 | 0.5587 | 0.8000 | 0.6435 | 0.6406 | 0.8654 | 0.5312 | 0.5285 | 0.7370 | 0.6674 | 0.6485 | 0.8393 | 0.3273 | 0.4622 | 0.8091 |
| LR Probe | 0.6167 | 0.4667 | 0.7500 | 0.7600 | 0.7683 | 0.6000 | 0.7772 | 0.7176 | 0.5754 | 0.7906 | 0.8344 | 0.7500 | 0.6821 | 0.6443 | 0.5768 | 0.7364 | 0.7493 | 0.6455 |
| SAPLMA | 0.6728 | 0.5006 | 0.8876 | 0.6800 | 0.6962 | 0.4402 | 0.7510 | 0.7240 | 0.5962 | 0.8438 | 0.8406 | 0.7572 | 0.6304 | 0.5880 | 0.7679 | 0.7909 | 0.7847 | 0.6327 |
| CoE | 0.6314 | 0.5108 | 0.7731 | 0.5400 | 0.6746 | 0.8576 | 0.7293 | 0.7740 | 0.7846 | 0.7500 | 0.8411 | 0.7542 | 0.6448 | 0.6526 | 0.8393 | 0.6545 | 0.5495 | 0.6954 |
| CoT-Kinetics | 0.6266 | 0.5824 | 0.7313 | 0.5800 | 0.6000 | 0.8500 | 0.7417 | 0.6000 | 0.7510 | 0.4219 | 0.5000 | 0.8500 | 0.3071 | 0.5000 | 0.8500 | 0.6455 | 0.6143 | 0.8394 |
| TRACED | 0.7088 | 0.6749 | 0.5397 | 0.7050 | 0.7328 | 0.4250 | 0.7825 | 0.7758 | 0.5700 | 0.8495 | 0.8422 | 0.7250 | 0.7194 | 0.6658 | 0.5375 | 0.7638 | 0.6333 | 0.6250 |
| *Llama-3.1-8B-Instruct* | | | | | | | | | | | | | | | | | | |
| MSP | 0.6237 | 0.6159 | 0.8786 | 0.3857 | 0.4917 | 0.8678 | 0.4770 | 0.5717 | 0.8821 | 0.5777 | 0.6196 | 0.8167 | 0.4817 | 0.5164 | 0.8913 | 0.4630 | 0.5230 | 0.8589 |
| Perplexity | 0.5839 | 0.5915 | 0.8677 | 0.3143 | 0.3709 | 0.8672 | 0.4483 | 0.4956 | 0.8546 | 0.5339 | 0.5197 | 0.8500 | 0.4933 | 0.5229 | 0.8744 | 0.5133 | 0.5468 | 0.8661 |
| Entropy | 0.6086 | 0.6065 | 0.8355 | 0.3186 | 0.3959 | 0.8324 | 0.4575 | 0.5276 | 0.8342 | 0.5822 | 0.5795 | 0.8333 | 0.4829 | 0.5220 | 0.8682 | 0.4719 | 0.5318 | 0.8615 |
| LR Probe | 0.5995 | 0.5567 | 0.7419 | 0.7571 | 0.6552 | 0.5000 | 0.6966 | 0.7199 | 0.8333 | 0.6362 | 0.6025 | 0.8667 | 0.6445 | 0.6225 | 0.8431 | 0.6435 | 0.6065 | 0.8077 |
| SAPLMA | 0.5720 | 0.5285 | 0.8710 | 0.7333 | 0.6661 | 0.4286 | 0.7011 | 0.7392 | 0.7767 | 0.6195 | 0.6046 | 0.8333 | 0.6640 | 0.6174 | 0.8608 | 0.6471 | 0.6551 | 0.7308 |
| CoE | 0.5516 | 0.5681 | 0.8677 | 0.3810 | 0.5403 | 0.8325 | 0.7471 | 0.7670 | 0.8643 | 0.5373 | 0.5193 | 0.8333 | 0.6233 | 0.5316 | 0.8324 | 0.4808 | 0.4835 | 0.8503 |
| CoT-Kinetics | 0.5269 | 0.4793 | 0.8535 | 0.5238 | 0.5399 | 0.8714 | 0.3276 | 0.4832 | 0.8534 | 0.4367 | 0.4991 | 0.8467 | 0.6236 | 0.5156 | 0.8437 | 0.5754 | 0.5000 | 0.8500 |
| TRACED | 0.6676 | 0.6290 | 0.6739 | 0.7344 | 0.6794 | 0.5250 | 0.7556 | 0.7714 | 0.7750 | 0.6363 | 0.6265 | 0.8000 | 0.7213 | 0.6233 | 0.7000 | 0.6550 | 0.6750 | 0.7250 |
| *Qwen2.5-7B-Instruct* | | | | | | | | | | | | | | | | | | |
| MSP | 0.4600 | 0.5457 | 0.8564 | 0.6389 | 0.6519 | 0.7778 | 0.6489 | 0.5659 | 0.8310 | 0.4828 | 0.5201 | 0.8833 | 0.6407 | 0.6069 | 0.8615 | 0.4583 | 0.4859 | 0.8375 |
| Perplexity | 0.4185 | 0.5178 | 0.8532 | 0.3917 | 0.3871 | 0.8723 | 0.3772 | 0.4123 | 0.8655 | 0.5567 | 0.6053 | 0.8317 | 0.5876 | 0.5796 | 0.8154 | 0.3958 | 0.5279 | 0.8375 |
| Entropy | 0.4385 | 0.5329 | 0.8375 | 0.6806 | 0.7038 | 0.7778 | 0.6495 | 0.5616 | 0.8136 | 0.4817 | 0.5200 | 0.8842 | 0.6492 | 0.6119 | 0.8463 | 0.4458 | 0.4725 | 0.8375 |
| LR Probe | 0.6109 | 0.7340 | 0.8125 | 0.7194 | 0.6680 | 0.5383 | 0.6721 | 0.6845 | 0.8138 | 0.7258 | 0.7349 | 0.8378 | 0.7081 | 0.6557 | 0.7750 | 0.7583 | 0.6957 | 0.8750 |
| SAPLMA | 0.6221 | 0.6818 | 0.8750 | 0.7622 | 0.7450 | 0.4257 | 0.6810 | 0.6870 | 0.8093 | 0.7003 | 0.7538 | 0.8384 | 0.7387 | 0.7013 | 0.7846 | 0.6958 | 0.7262 | 0.8375 |
| CoE | 0.5283 | 0.5724 | 0.8688 | 0.6833 | 0.6450 | 0.6889 | 0.6236 | 0.4476 | 0.8483 | 0.5586 | 0.6308 | 0.8863 | 0.7486 | 0.7362 | 0.7742 | 0.6625 | 0.6242 | 0.7015 |
| CoT-Kinetics | 0.5674 | 0.6079 | 0.8311 | 0.4722 | 0.6000 | 0.7444 | 0.6829 | 0.6877 | 0.8600 | 0.5442 | 0.5223 | 0.8386 | 0.7080 | 0.6218 | 0.8121 | 0.4417 | 0.5000 | 0.8906 |
| TRACED | 0.6238 | 0.7380 | 0.8060 | 0.7636 | 0.7583 | 0.4750 | 0.6956 | 0.6895 | 0.8024 | 0.7305 | 0.7859 | 0.8280 | 0.7794 | 0.7371 | 0.7733 | 0.7752 | 0.7314 | 0.6125 |

the Central Limit Theorem (Feller, 1991), as the metrics $M$ and $K$ are cumulative aggregations of step-wise dynamics that naturally converge to normality.

**Prior** $P(y_n = c)$**:** As the prior distribution approaches a uniform balance $P(y_n = 1) = P(y_n = 0)$, the influence of external distributional biases diminishes, naturally grounding the assessment on intrinsic geometric evidence. However, empirical validation confirms that TRACED maintains robust stability even under a certain prior imbalances (see Appendix H).

**Decision Rule.** The model classifies the reasoning quality by selecting the class with the higher posterior probability:

$$\hat{y}_n = \mathbb{I}\left[\log \frac{P(y_n = 1)P(\mathbf{x}_n|y_n = 1)}{P(y_n = 0)P(\mathbf{x}_n|y_n = 0)} > 0\right] \quad (7)$$

This framework is adaptive and avoids manual threshold search. We do not manually set specific classification thresholds. Instead, the framework learns the natural geometric boundaries of good reasoning directly from manifold topology, automatically adjusting to different reasoning tasks.

## 4. Experiments

### 4.1. Experimental Setup

**Datasets and Metrics.** We evaluate our framework across six benchmarks spanning two distinct domains: 1) **Structured Reasoning**, requiring strict deduction, covers mathematics (GSM8K (Cobbe et al., 2021), MATH (Hendrycks et al., 2021)), theorem proving (TheoremQA (Chen et al., 2023)), and scientific reasoning (GPQA (Rein et al., 2024)); 2) **Open-Ended Reasoning**, necessitating divergent thinking, includes Social IQA (Sap et al., 2019) (social dynamics) and Understanding Fables (Srivastava et al., 2023) (moral abstraction). Performance is assessed via standard binary classification metrics: **AUROC** (Boyd et al., 2013), **AUPR**, and **FPR@95** (Manning & Schütze, 1999). **Dataset construction, spliting and labeling** details are provided in Appendix A and Table 6.

**Models and Baselines.** We employ two Instruction-Tuned models (*Qwen2.5-7B-Instruct* (Team et al., 2024), *Llama-3.1-8B-Instruct* (Grattafiori et al., 2024)) and two Reasoning models (*DeepSeek-R1-Llama-8B* (Guo et al., 2025), *Qwen3-4B-Thinking-2507* (Yang et al., 2025a)). We compare against three established baseline categories: (1) **Output Probability Methods** using scalar statistics (e.g., *MSP*, *Perplexity* (Si et al., 2022)); (2) **Hidden State Probes** based on supervised linear classifiers (*LR Probe* (Alain & Bengio, 2017), *SAPLMA* (Azaria & Mitchell, 2023)); and (3) **Trajectory Modeling Methods** utilizing geometric features (*CoE* (Wang et al., 2024), *CoT-Kinetics* (Bi et al., 2025)). See Appendix C for more details.

### 4.2. Main Results

**TRACED excels in structured reasoning tasks, demonstrating robust evaluation performance across diverse model architectures.** As shown in Table 1, TRACED con-

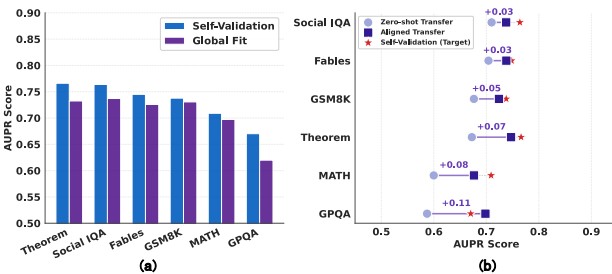

*Figure 2.* **Universality and Generalization Analysis. (a) Universal Signature:** A single *global fit model* derived from aggregated data achieves competitive AUPR across diverse tasks, supporting the existence of a task-agnostic geometric signature. **(b) Cross-Domain Adaptation:** Dumbbell plot comparing *Direct Zero-shot Transfer* (blue circles), *Aligned Transfer* (purple squares), and *Supervised In-domain Upper Bound* (red stars). Results confirm that the geometric alignment significantly bridges the performance gap caused by distribution shifts.

*Table 2.* **Robustness of TRACED Across Reasoning Complexity.** Performance of *DeepSeek-R1-Llama-8B* stratified by reasoning steps ($L$). **Gap ($\Delta$)** denotes the maximum fluctuation across difficulty tiers. The low variance ($\Delta \leq 2.7\%$) confirms stability. Comprehensive results for all models are provided in Table 10.

| Metric | Easy ($L \leq 4$) | Medium ($5 \leq L \leq 8$) | Hard ($L > 8$) | Gap ($\Delta$) |
|---|---|---|---|---|
| AUROC ($\uparrow$) | 0.775 | 0.748 | 0.766 | 2.7% |
| AUPR ($\uparrow$) | 0.708 | 0.710 | 0.723 | 1.5% |
| FPR@95 ($\downarrow$) | 0.660 | 0.673 | 0.685 | 2.5% |

*Figure 3.* **Robustness and Efficiency. (Left) Class Imbalance:** TRACED maintains discriminative stability against distributional shifts, specifically where the prior $P(y_n = 1) \in [0.3, 0.7]$. **(Right) Data Efficiency:** The method achieves rapid geometric convergence, reaching a stability plateau with merely $N \approx 400$ reference samples.

sistently outperforms standard *Output Probability Methods* (e.g., MSP, Perplexity) across all models, demonstrating that geometric features provide a richer correctness signal than scalar probabilities. Against supervised *Hidden State Probes*, TRACED remains highly competitive on benchmarks like GSM8K and MATH, suggesting that modeling the geometric evolution of the entire reasoning process yields a more holistic assessment than classifiers restricted to the final token. While parametric supervised methods marginally lead in specific configurations (e.g., TheoremQA), TRACED consistently secures the second-best performance and matches supervised baselines on challenging tasks like GPQA. Furthermore, TRACED surpasses prior *Trajectory Modeling Methods*, indicating that temporal geometric features characterize the cognitive process more effectively than layer-wise measures (CoE) or dynamical equation modeling (CoT-Kinetics).

**TRACED demonstrates exceptional performance on divergent, open-ended reasoning tasks.** By modeling the geometric evolution of the reasoning process, our method captures the nuances of divergent thinking, consistently outperforming baselines on tasks requiring deep contextual understanding (e.g., Social IQA, Fables). Notably, performance shift compared to structured domains: TRACED and other Trajectory Modeling Methods frequently surpass supervised Hidden State Probes in these tasks (e.g., CoE outperforms LR probe on SocialIQA with Qwen2.5-7B-Instruct). This corroborates that the complexity of divergent thinking renders static final-token representations insufficient, necessitating the integration of information accumulated throughout the entire reasoning trajectory.

**Universality and Cross-Domain Robustness of Geometric Signatures.** We posit that reasoning quality is encoded in the intrinsic topology of the representation manifold (i.e., displacement and curvature) rather than task-specific se-

mantics, forming a domain-invariant signal. To validate this, we compared a *Global Fit* model (derived from aggregated data) against task-specific *Self-Validation* baselines (in-domain upper bounds). As shown in Figure 2(a), although the Global Fit model naturally lags slightly behind specialized oracles, it retains substantial performance across most domains, achieving competitive AUPR scores without task-specific fine-tuning.

While the universal signature generalizes well, a performance gap persists compared to in-domain upper bounds, likely due to distributional shifts in feature magnitude (e.g., smaller displacements in scientific reasoning versus narratives) rather than intrinsic feature failure. We address this via **centroid alignment**, adapting the source $\mathcal{D}_S$ to the unseen target $\mathcal{D}_T$ through rigid translation ($\Delta\boldsymbol{\mu} = \boldsymbol{\mu}_T - \boldsymbol{\mu}_S$). As shown in Figure 2(b), this alignment yields substantial recovery, confirming that the drop stemmed solely from distributional misalignment. These findings suggest that while tasks occupy different absolute regions in latent space, their *relative topological structure* regarding quality remains isomorphic. We demonstrate that TRACED shows superior deployment efficiency and robust cross-domain transferability, as detailed in Appendix J.

**Robustness Across Reasoning Complexity.** We evaluate TRACED's stability against problem difficulty, quantifying complexity by the number of essential reasoning steps ($L$) (Shojaee et al., 2025; Zhao et al., 2025). Using stratified uniform sampling across all domains, we categorize samples into three tiers: *Easy* ($L \leq 4$), *Medium* ($5 \leq L \leq 8$), and *Hard* ($L > 8$) (details in Appendix L). As shown in Table 10, TRACED maintains consistent detection capability as complexity escalates, with performance fluctuations remaining minimal ($\Delta \leq 2.8\%$) across all models. This confirms that our method effectively captures trajectory quality

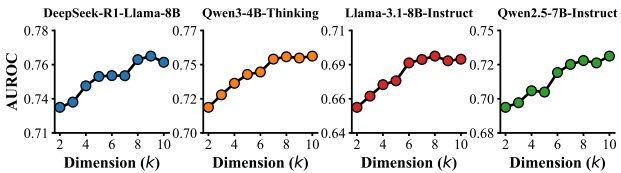

*Figure 4.* **Sensitivity to Subspace Dimension** $k$. AUROC evaluation across four models ($k \in [2, 10]$) shows performance improves and stabilizes at $k = 8$. Additional metrics (AUPR, FPR@95) are detailed in Appendix M.

*Table 3.* **Component Ablation.** Performance comparison of individual geometric signatures ($M_n$ only, $K_n$ only) versus the full TRACED framework (See Appendix G for more results).

| Config | $M_n$ | $K_n$ | Fables | GPQA | GSM8K | MATH | Soc_IQA | ThrmQA |
|---|---|---|---|---|---|---|---|---|
| Disp. Only | ✓ | | 0.6845 | 0.7812 | 0.7634 | 0.7012 | 0.7122 | 0.8245 |
| Curv. Only | | ✓ | 0.6512 | 0.7244 | 0.7188 | 0.6855 | 0.6945 | 0.7912 |
| **TRACED** | ✓ | ✓ | **0.7191** | **0.8300** | **0.8061** | **0.7489** | **0.7536** | **0.8730** |

independent of reasoning complexity and length.

**Robustness and Efficiency.** As visualized in Figure 3, TRACED retains **robust discriminative power** against moderate prior mismatches ($\alpha \in [0.3, 0.7]$, Fig. 3a), and demonstrates **high data efficiency**, reaching distributional stability with limited samples ($N \approx 400$, Fig. 3b). See Appendix H for extended analysis and results.

**Component Ablation and Hyperparameter Sensitivity.** (1) Regarding the subspace dimension $k$ of $B$ (Step 5 in Algorithm 1), Figure 4 shows the performance remains robust and converges at $k = 8$, demonstrating that a low-rank subspace sufficiently encodes the essential kinematic signals. (2) Moreover, component ablation analysis (Table 3) confirms the synergy of Displacement ($M_n$) and Curvature ($K_n$), where their integration consistently yields superior discriminative power over individual features.

**The Efficacy of Subspace Projection and Geometric Kinematics.** We evaluate the necessity of the reasoning quality subspace by comparing the full TRACED framework against a "No-Subspace" variant, as shown in Table 4. The full framework consistently achieves superior AUPR scores across tasks, demonstrating that the subspace basis effectively filters out non-logical artifacts such as syntax and formatting. Crucially, even without subspace projection, the variant relying solely on raw geometric features (displacement and curvature) still outperforms traditional baselines like MSP. This validates the exceptional discriminative power inherent in the underlying physical mechanism of abstracting the reasoning process as geometric trajectories within the latent space.

### 4.3. Kinematic Scaling Laws of Reasoning

To empirically validate the theory of kinematic regimes in Appendix E, we focus on the **Net Displacement** $D(T) = \|\mathbf{z}_T - \mathbf{z}_0\|_2$ as a function of reasoning length $T$ (token count).

*Table 4.* No-Subspace Ablation.

| Model | Strategy | GPQA | GSM8K |
|---|---|---|---|
| DeepSeek-R1-Llama-8B | No-Subspace | 0.624 | 0.813 |
| | **TRACED** | **0.661** | **0.828** |
| Qwen2.5-4B | No-Subspace | 0.712 | 0.752 |
| | **TRACED** | **0.733** | **0.776** |

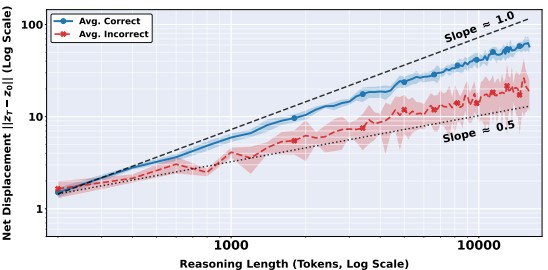

*Figure 5.* **Kinematic Scaling Laws of Reasoning.** Log-log plot of Net Displacement $D(t) = \|z_T - z_0\|_2$ vs. reasoning length across six domains. **Blue (Correct):** Exhibits linear scaling ($slope \approx 0.82$), characteristic of directed evolution ($D \propto T$) where computation yields direct semantic progress. **Red (Incorrect):** Follows sub-linear scaling ($slope \approx 0.53$), resembling random walk ($D \propto \sqrt{T}$) and indicating progress stagnation. Shaded regions denote standard deviation.

Based on this metric, the **theory of kinematic regimes** postulates that reasoning dynamics bifurcate into two distinct asymptotic behaviors: *Correct Reasoning* follows linear scaling ($D(T) \propto T$), whereas *Incorrect Reasoning* exhibits sub-linear scaling ($D(T) \propto T^{0.5}$). We analyze the average geometric evolution across diverse domains (setup details in Appendix F), and Figure 5 confirms this distinct topological phase transition. Correct reasoning (blue) adheres to directed linear scaling ($slope \approx 0.82$), indicating that computational steps translate proportionally into semantic progress ($O(t)$). In contrast, Incorrect chains (red) follow sub-linear scaling ($slope \approx 0.53$), revealing that low-quality reasoning behaves as a random walk confined in the semantic space. This provides a **fundamental kinematic explanation** for our earlier observation regarding the significant divergence in accumulated displacement ($M_n$) between correct and incorrect reasoning.

### 4.4. Geometric Differences Across Domains

Beyond differentiating correctness, we investigate whether the correct geometry of thought varies across reasoning domain. Figure 6 reveals how reasoning patterns differ fundamentally between *Structured Domains* and *Open-Ended Domains* (detailed setup in Appendix K).

**Curvature (Left):** Structured reasoning exhibits a highly concentrated distribution, reflecting "Logical Stiffness", where any minor semantic deviation (increased curvature) risks breaking the logical chain. In contrast, open-ended reasoning follows a long-tailed distribution, indicating that

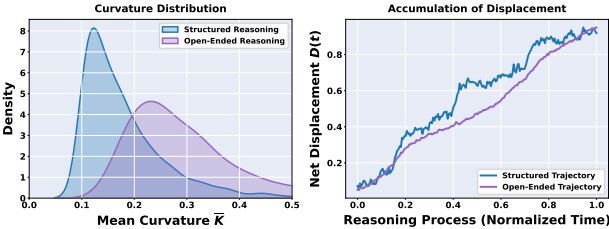

*Figure 6.* **Geometric Differences Across Domains. (Left) Curvature Distribution:** Structured reasoning (blue) exhibits a narrow peak, contrasting with the broad, heavy tail of open-ended reasoning (purple). **(Right) Displacement Accumulation:** Structured trajectories reveal step-wise growth driven by discrete breakthroughs, while open-ended tasks exhibit a smooth, continuous semantic flow.

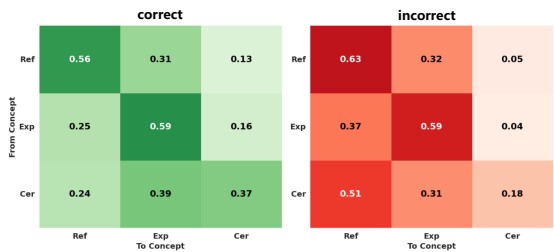

*Figure 7.* **Transition Structures of Cognitive States.** Visualization of transition probabilities $P(S_{t+1}|S_t)$ of Correct and Incorrect Reasoning.

open contexts permit and even encourage a degree of thought dispersion, provided the core narrative remains intact. **Displacement (Right):** Structured tasks exhibit **step-wise transitions**, where solving a critical sub-problem triggers a sudden jump in semantic distance. Conversely, open-ended tasks follow a **smooth gradient**, reflecting a steady accumulation of narrative understanding that gradually saturates as the description deepens.

### 4.5. Cognitive State Dynamics

To investigate the dynamics of cognitive states during reasoning, we model the transition dynamics between cognitive concept states (Reflection, Exploration, Certainty), regarded as different stages of reasoning by previous studies (Chen et al., 2025a). See Appendix D for concept extraction details. Figure 7 illustrates the transition probabilities $P(S_{t+1}|S_t)$, revealing that high-quality reasoning converges effectively toward Certainty, whereas low-quality reasoning remains trapped in loops that fail to reach a certainty conclusion.

**1) Accessibility of Certainty.** A primary distinction is the accessibility of Certainty. In correct reasoning, both *Reflection* and *Exploration* drive directed progression to *Certainty* ($P \approx 0.13$ and $0.16$). Conversely, incorrect trajectories face a transition bottleneck (probabilities collapse to $< 0.05$), preventing the model from reaching a stable conclusive state. **2) Hesitation Loops.** Incorrect reasoning exhibits higher regression from *Exploration* back to *Reflection* (0.37 vs. 0.25). This indicates logical dead-ends that force the model to retreat to earlier phases, resulting in an **oscillatory**

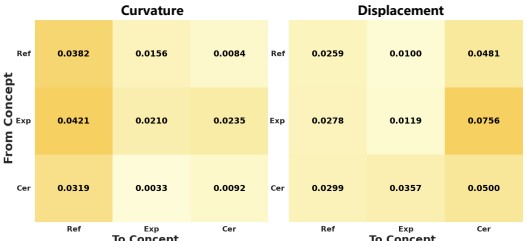

*Figure 8.* **Geometric Cost of State Transitions.** (Left) Avg. curvature change ($\Delta K$). (Right) Avg. displacement change ($\Delta M$). Curvature encodes the cost of uncertain directional reorientation, while displacement reflects accumulative semantic progress.

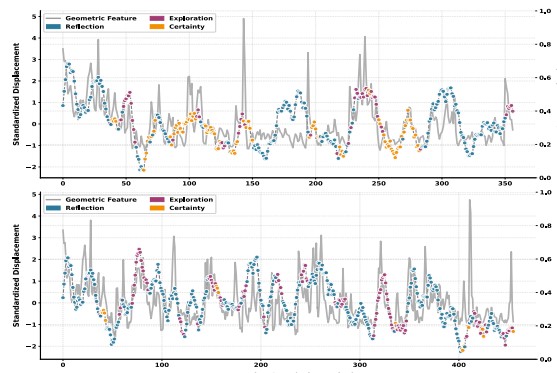

*Figure 9.* **Geometric-Semantic Synchronization.** Alignment between geometric displacement (gray) and cognitive states.

**pattern** where it cycles fruitlessly between hesitation and exploration without forward progress. **3) Stability and Persistence.** Correct reasoning demonstrates strong **temporal persistence** in *Certainty* ($P(Cer|Cer) \approx 0.37$ vs. 0.18), implying the model maintains confidence long enough to consolidate thoughts. In contrast, incorrect reasoning suffers from **structural instability**, where *Certainty* frequently reverts to *Reflection* ($P(Ref|Cer) \approx 0.51$, double the correct rate), signaling premature loss of confidence.

### 4.6. The Bridge: Linking Semantic to Geometry

To ground the cognitive semantics (in Sec. 4.5) in geometric properties, we analyze the *geometric cost* associated with each state transition $S_i \rightarrow S_j$, as shown in Figure 8.

**1) Curvature as the Cost of Uncertainty.** The heatmap identifies cognitive reorientation as the primary driver of directional instability. Peak curvature occurs during regressions from exploration to reflection ($Exp \rightarrow Ref, \Delta K \approx 0.042$), confirming that "Hesitation Loops" demand significant representational shifts. Similarly, abandoning certainty ($Cer \rightarrow Ref, \Delta K \approx 0.031$) induces sharp geometric turns. These frequent reorientations disrupt trajectory stability, explaining the high-curvature profile of incorrect reasoning.

**2) Displacement as Accumulative Progress.** Conversely, displacement is governed by the convergence and maintenance of Certainty. Transitions into and within certainty

*Table 5.* Alignment Training Results. Performance comparison of Vanilla, standard GRPO, and TRACED-GRPO on GSM8K and MATH benchmarks.

| Model | Method | Training Configuration | | | | Evaluation Results | |
|---|---|---|---|---|---|---|---|
| | | Learning Rate | Group ($G$) | KL $\beta$ | TRACED $\lambda$ | GSM8K (%) | MATH (%) |
| Qwen2.5-0.5B | Vanilla | - | - | - | - | 30.93 | 24.60 |
| | GRPO | 3e-6 | 8 | 0.04 | - | 39.65 | 28.33 |
| | **TRACED-GRPO** | 3e-6 | 8 | 0.04 | 0.1 | **43.97** | **30.65** |
| Qwen2.5-1.5B | Vanilla | - | - | - | - | 62.25 | 44.00 |
| | GRPO | 1e-6 | 8 | 0.02 | - | 67.83 | 47.20 |
| | **TRACED-GRPO** | 1e-6 | 8 | 0.02 | 0.1 | **71.22** | **50.10** |

\* The 1.5B model employs LoRA fine-tuning ($r = 8, \alpha = 16$); the 0.5B model uses full-parameter fine-tuning.

yield maximum semantic distance ($Exp \to Cer, \Delta M \approx 0.076; Cer \to Cer, \Delta M \approx 0.050$). This confirms displacement as a proxy for semantic progress: correct reasoning maximizes net movement by sustaining confidence phases.

**3) Empirical Validation of Synchronization.** Figure 9 maps displacement to cognitive states. *Correct reasoning* (top) exhibits sustained Certainty alongside displacement peaks, establishing displacement as a physical manifestation of confidence. In contrast, *incorrect reasoning* (bottom) suffers from Exploration-Reflection oscillations. This semantic turbulence mirrors the geometric "Hesitation Loop," where the inability to converge to Certainty halts displacement accumulation. See Appendix O for qualitative visualizations of these reasoning dynamics.

### 4.7. Extension: TRACED as a Reliable Reward Signal

**The Potential of TRACED as a Reliable Reward Signal.** We integrate the geometric kinematics score, $S_{\text{trace}} = \text{Norm}(M_n) - \text{Norm}(K_n)$, into the GRPO framework, formulating the final reward as $R_{\text{final}} = R_{\text{ans}} + \lambda \cdot S_{\text{trace}}$ (where $\lambda = 0.1$). Here, $R_{\text{ans}}$ denotes the standard GRPO reward, which explicitly consists of the outcome accuracy reward and the process formatting reward. We conducted preliminary alignment training experiments on the Qwen2.5-0.5B and Qwen2.5-1.5B models, using a training set that covers seven mathematical sub-disciplines scientifically proportioned across different difficulty levels. As demonstrated in Table 5, experimental results demonstrate that TRACED-GRPO outperforms the traditional Vanilla GRPO on mathematical benchmark tasks such as GSM8K and MATH. This performance stems from two core mechanisms.

First, TRACED effectively mitigates the issue of zero reward variance caused by all samples in a batch being incorrect in high-difficulty tasks. By assigning non-parametric geometric scores based on trajectory quality, it provides a crucial tie-breaking signal when outcome-based rewards fail, ensuring that advantage values do not collapse to zero, thereby maintaining stable gradient updates. Second, by strictly penalizing high-curvature hesitation loops and rewarding net semantic displacement, TRACED alleviates reward hacking behaviors caused by sparse outcome-based rewards. It restrains, to a certain extent, the false alignment phenomenon where models accidentally achieve high

scores through incorrect or redundant logical paths, thereby forcefully guiding the policy model to converge toward a structurally rigorous, logically concise, and robust authentic reasoning paradigm.

## 5. Related Works

**Assessment of Reasoning Quality.** Existing methods range from **resource-intensive** external verifiers (Xiong et al., 2023; Li et al., 2022; Zhang et al., 2025c) to **performance-limited** intrinsic probability metrics (Huang et al., 2023; He et al., 2025). While some methods analyze hidden state evolution (Wang et al., 2024; Bi et al., 2025), they typically neglect critical temporal signals by modeling averaged token representations. Unlike prior works, we construct an evaluation signal based on theoretically grounded geometric features of temporal reasoning, achieving consistent improvements and scalability across diverse tasks.

**Representation Analysis of Reasoning.** Research on internal representations has expanded to temporal dimensions and geometric metrics (Vilas et al., 2025; Li et al., 2025; Wang et al., 2024; Hu et al., 2024; Zhang et al., 2025b). However, prior works often lack explicit geometric interpretability or fail to explain the correspondence between reasoning behaviors and geometric variations (Zhou et al., 2025; Manson, 2025). In contrast, we uncover the intrinsic correspondence between geometric formalism and cognitive reasoning behavior, advancing the interpretability of the reasoning process. See details in Appendix P.

## 6. Conclusion

We introduce TRACED, a framework that evaluates LLM reasoning via geometric kinematics. By quantifying the Progress and Stability, we reveal that correct reasoning manifests as high-progress, stable trajectories, whereas incorrect are characterized by low-progress, unstable patterns. This topological distinction enables competitive performance and superior robustness across diverse benchmarks. We also bridges geometry and cognition by interpreting curvature as "Hesitation Loops" and displacement as Certainty, providing a physical lens to decode machine thought.

### Acknowledgement
This work are partially supported by the funding BAS/1/1689-01-01, RGC/3/7125-01-01, FCC/1/5940-20-05, FCC/1/5940-06-02, and King Abdullah University of Science and Technology (KAUST) – Center of Excellence for Generative AI, under award number 5940 and a gift from Google, and MBZUAI Research Fund BF0100.

### Impact Statement

This paper presents work whose goal is to advance the field of Machine Learning. There are many potential societal consequences of our work, none of which we feel must be specifically highlighted here.

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

# A. Dataset Construction and Labeling Details

To construct a robust dataset of paired correct and incorrect reasoning traces, we followed a standardized generation and verification pipeline.

## A.1. Reasoning Trajectory Generation

We generated $N = 10$ distinct reasoning paths for each question to construct the geometric manifold. The generation configuration is strictly tailored to the model architecture and dataset characteristics.

**1. Generation Hyperparameters. Temperature Sampling** ($T = 0.7$)**:** We set the temperature to **0.7** for all generation. This setting strikes an optimal balance: it introduces sufficient stochasticity to reveal diverse reasoning paths (and potential hallucinations) for topological analysis, while maintaining enough coherence.

**Maximum Token Limit:** (1)**Standard Instruct Models:** Set to **4,096 tokens** to cover standard chain-of-thought derivations. (2)**Large Reasoning Models (LRMs):** For LRMs, we extended the limit to **16,384 tokens** to prevent the truncation of the extensive internal `<think>` phase.

**Leakage Prevention via Question-Level Splitting.** To rigorously prevent data leakage and ensure evaluation fairness, we enforce a strict separation based on unique question prompts. Specifically, the dataset partition is performed on the question level rather than the trajectory level. Consequently, all $N$ trajectories derived from a specific question reside exclusively in either the calibration set or the evaluation set. This guarantees that the model never encounters reasoning traces from the same prompt during the calibration phase, ensuring zero overlap between splits.

**2. Task-Specific Prompting.** To ensure validity, we adopted task-specific templates derived from OpenCompass. We classify our benchmarks into four structural categories based on the required output format (see Table 6 for dataset mapping).

*Table 6.* **Overview of the Datasets.** Statistics of the original source datasets. **Note:** Due to the cost of constructing reasoning trajectories ($N = 10$ trajectories per sample), we randomly sampled 1,000 instances for experimentation. All collected samples were subsequently partitioned into training and evaluation sets using a randomized 8:2 split.

| Dataset | Reference | Task Description | Metric | Format | Prompt | Example (Truncated) |
|---|---|---|---|---|---|---|
| GSM8K | (Cobbe et al., 2021) | Multi-step grade school mathematics problems. | Acc. | Number | Type A | *Q: Natalia sold clips to 48 friends... A: 48* |
| MATH | (Hendrycks et al., 2021) | Challenging competition-level math. | Acc. | LaTeX | Type C | *Q: Find $f(f(2))$. A: \boxed{298}* |
| TheoremQA | (Chen et al., 2023) | Application of STEM theorems to solve novel problems. | Acc. | Option / Boolean / Number | Type D | *Q: Calculate derivative of $x^2$ at 3. A: 6.0* |
| GPQA | (Rein et al., 2024) | PhD-level scientific QA. | Acc. | Option | Type B | *Q: Which mechanism drives... A: (B)* |
| Social IQA | (Sap et al., 2019) | Reasoning about social interactions and motivations. | Acc. | Option | Type B | *Q: Tracy pressed wrong button... A: (A)* |
| Fables | (Srivastava et al., 2023) | Abstracting morals from allegorical narratives. | Acc. | Option | Type B | *Q: The Tortoise and the Hare implies... A: (A)* |

## A.2. Ground Truth Verification

To ensure the high fidelity of our reasoning quality labels, we implemented a rigorous Two-Pronged Verification Strategy. This pipeline integrates strict programmatic parsing with an LLM-based semantic judge to label generated trajectories as *Positive* (Correct) or *Negative* (Incorrect).

### A.2.1. STAGE 1: DETERMINISTIC PROGRAMMATIC PARSING

We first employ task-specific parsers to extract and normalize the model's output. To ensure data integrity, any trajectory missing End-of-Sequence tokens (e.g., `<|im_end|>`, ``) is considered incomplete and is **excluded from the dataset**. For completed traces, we apply the following logic based on the dataset type:

---

**Task-Specific Instructions**

**Type A: Math Word Problems (GSM8K)**
*Output Requirement: A specific numerical value.*

```
Answer the following math problem.  [CoT Trigger].  The last line of your response
should be of the following format:  "Answer: <NUMBER>" (without quotes), where
<NUMBER> is the final calculated value. Question: {input_data}

Response:
```

**Type B: Multiple Choice Reasoning (GPQA, Social IQA, Understanding Fables)**
*Output Requirement: Selection of a specific option letter (A, B, C, D).*

```
Answer the following multiple-choice question.  [CoT Trigger].  The last line of
your response should be of the following format:  "Answer: <LETTER>" (without
quotes), where <LETTER> is one of the provided options (e.g., A, B, C, D).  Question:
{input_data}
Response:
```

**Type C: Advanced Mathematics (MATH)**
*Output Requirement: LaTeX formatted expression.*

```
Question:  {input_data}
Please [CoT Trigger], and put your final answer within \boxed{}.
Response:
```

**Type D: Theorem Proving (TheoremQA)**
*Output Requirement: Dynamic types (Boolean, List, Number, etc.).*

```
Instruction:
Please read a math problem.  [CoT Trigger].  The answer is decided by Answer Type.
If the Answer type in [bool], the answer needs to be True or False.
Else if the Answer type in [integer, float], The answer needs to be in numerical
form.
Else if the Answer type in [list of integer, list of float], the answer needs to be
a list of number like [2, 3, 4].
Else if the Answer type in [option], the answer needs to be an option like (a), (b),
(c), (d).
You need to output the answer in your final sentence like 'Therefore, the answer is
...'.
 Question:  {input_data}
Answer_type:  {answer_type}
Response:
```

*Note on CoT Triggers:* For Standard Instruct Models, `[CoT Trigger]` is replaced with "Think step by step". For LRMs (e.g., DeepSeek-R1), this trigger is omitted to respect the model's native reinforcement-learned thinking patterns, relying solely on format constraints.

1. **Type A: Numeric Extraction (GSM8K).** For arithmetic tasks, we employ a regex-based extraction pipeline robust to formatting noise. We verify if the specific `answer_prefix` (e.g., "Answer:") exists. If found, we extract the subsequent text and identify all numeric values using the regex `r"\d+\.?\d*"`. We select the **last identified number** as the prediction. Both the prediction and ground truth are normalized by removing commas (thousands separators) and trailing zeros/decimals (e.g., converting "1,000.0" to "1000") before performing an exact equivalence check.

2. **Type B: Multiple Choice Matching (GPQA, Social IQA, Fables).** For option-selection tasks, we implement a parser to identify the predicted option letter. The parser scans the text following the answer delimiter for patterns matching `r"(?i)Answer:\s*?([A-D])?"`. We extract the final matching capture group, normalize it to uppercase, and perform an exact string match against the ground truth letter.

3. **Type C: Symbolic Matching (MATH).** For complex mathematical expressions, we rely on the LaTeX `\boxed{}` format. We extract the string content within the last `\boxed{}` tag. To handle variability in LaTeX spacing, we normalize both the extracted content and the gold label by stripping all whitespace (e.g., $x + y \rightarrow x+y$) before comparison.

4. **Type D: Dynamic Extraction (TheoremQA).** Given the heterogeneous output formats of TheoremQA, we implement a conditional parser guided by the sample's `Answer_type`. We first isolate the concluding segment following the phrase "answer is".

   - **Bool:** We scan for case-insensitive occurrences of "True" or "False".

   - **Integer/Float:** We apply the same regex extraction strategy as Type A to retrieve the final numerical value.

   - **List:** We extract content enclosed within square brackets using the regex `r"\[(.*?)\]"`.

   - **Option:** We identify parenthesized characters (e.g., `(a)`) using the regex `r"\(([a-d])\)"`.

   The extracted prediction is then compared against the ground truth, which is similarly parsed to match the expected format (e.g., stripping brackets for lists).

### A.2.2. STAGE 2: LLM-AS-A-JUDGE VERIFICATION

Sole reliance on string matching can yield False Negatives due to semantic equivalence (e.g., $\frac{1}{2}$ vs. $0.5$) or rigid formatting requirements. To mitigate this, we employ a powerful instruction-tuned model (Llama-3-70B-Instruct (Dubey et al., 2024)) as an expert judge for cases where programmatic parsing yields a mismatch or ambiguity. We construct a verification prompt containing the *Question*, *Gold Answer*, and *Model Generation*. The judge is instructed to evaluate the semantic correctness of the reasoning chain and final answer, outputting a binary `CORRECT` or `INCORRECT` verdict.

### A.2.3. RELIABILITY ANALYSIS AND BIAS MITIGATION

To address concerns regarding potential bias from the LLM judge, we implemented the following audit and mitigation measures:

**1. Minimal Dependence:** The majority of samples are labeled by the deterministic parser, with the LLM judge required only for the remaining ambiguous instances. This limits the influence of any potential model bias to a small fraction of the dataset.

**2. Judge Audit:** To verify the reliability of the semantic judge, we performed a manual audit on a random subset of 100 trajectories labeled by the judge. We observed an agreement rate of over 95% with human annotation, confirming that the instruction-tuned Llama-3-70B provides high-fidelity verdicts for these objective reasoning tasks.

### A.2.4. FINAL LABELING PROTOCOL

A trajectory is labeled as Correct if it satisfies both the strict programmatic matching criteria and receives a positive verdict from the LLM Judge. This hybrid approach ensures we capture correct reasoning, providing a robust dataset for geometric analysis.

## B. Additional Geometric Visualizations across Models

To demonstrate that the geometric signatures of reasoning quality (Displacement and Curvature) are robust to model architectures, we provide additional visualizations for three other state-of-the-art LLMs: **Qwen2.5-7B-Instruct**, **Llama-3.1-8B-Instruct**, and **Qwen3-4B-Thinking-2507**.

As shown in Figures 10a, 10b, and 10c, the topological separation observed in the main text (Figure 1) is universally consistent. Across all models, correct reasoning traces (blue) are characterized by high displacement and low curvature, indicating effective semantic progress and logical stability. In contrast, incorrect traces (red) consistently exhibit the "Hesitation Loop" pattern (high curvature and low displacement) confirming that these geometric features are indicators of reasoning validity rather than model-specific artifacts.

## C. Baseline Implementation Details

In this section, we provide detailed implementation specifications for the baseline methods used in our comparative evaluation.

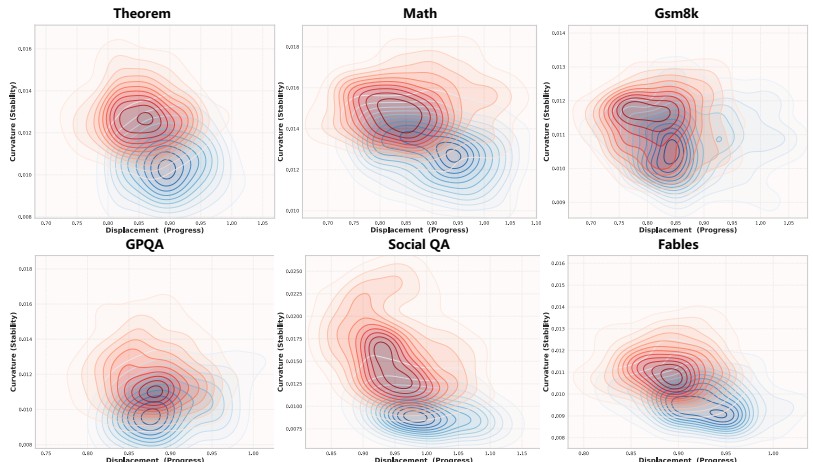

*(a)* **Qwen2.5-7B-Instruct.** The joint distribution of displacement ($M$) and curvature ($K$) confirms the topological separation between correct and incorrect reasoning.

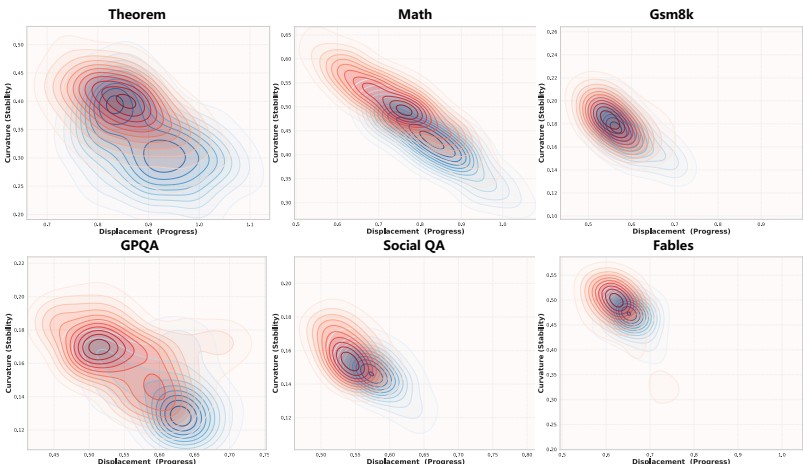

*(b)* **Llama-3.1-8B-Instruct.** Similar to other models, Llama-3.1 exhibits a distinct separation where high-quality reasoning maximizes displacement while minimizing curvature.

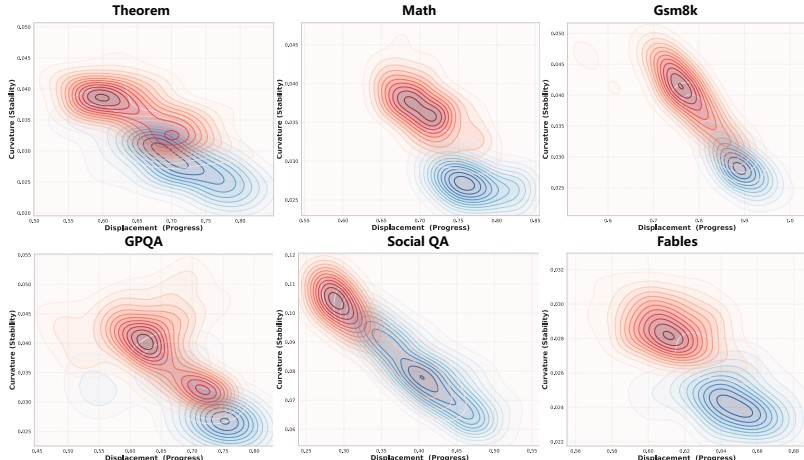

*(c)* **Qwen3-4B-Thinking-2507.** Even for models specialized in thinking processes, the geometric distinction remains robust: hallucinations manifest as high-curvature oscillations.

*Figure 10.* **Geometric Modes Across Different Models.** Visualization of the reasoning geometric signatures for Qwen2.5, Llama-3.1, and Qwen3-Thinking. Consistent topological separation is observed across all architectures.

## C.1. Output Probability Methods

These unsupervised methods rely on scalar probability distributions output by the model, ignoring vector-space structures.

**LN-Entropy (Length-Normalized Entropy) (Malinin & Gales, 2020).** Calculates the average predictive entropy over the sequence to measure uncertainty, normalized by token length to mitigate verbosity bias.

$$\mathcal{S}_{\text{Ent}} = \frac{1}{T} \sum_{t=1}^{T} \mathcal{H}(P(x_t|x_{<t})) \tag{8}$$

**MSP (Maximum Softmax Probability).** Measures confidence by tracking the average log-probability of the generated tokens.

$$\mathcal{S}_{\text{MSP}} = \frac{1}{T} \sum_{t=1}^{T} \log P(\hat{x}_t|x_{<t}) \tag{9}$$

**Perplexity (PPL) (Si et al., 2022).** Computes the exponentiated average negative log-likelihood, representing how "surprised" the model is by its own generation. Lower perplexity suggests higher fluency and confidence.

## C.2. Hidden State Probes

These methods train supervised classifiers on internal activations to distinguish correct from incorrect reasoning.

**LR Probe (Linear Regression Probe) (Alain & Bengio, 2017).** Trains a linear logistic regression classifier on the static hidden state of the final token $\mathbf{h}_T$. It tests whether reasoning correctness is linearly separable in the representation space.

**SAPLMA (Azaria & Mitchell, 2023).** Utilizes a non-linear Multi-Layer Perceptron (MLP) classifier trained on hidden layer activations. Unlike linear probes, SAPLMA learns complex, non-linear decision boundaries to predict the "truthfulness" of a statement directly from the model's internal belief state.

## C.3. Trajectory Dynamics Methods

These methods analyze the geometric or physical evolution of hidden states across layers.

**Chain of Embedding (CoE) (Wang et al., 2024).** Evaluates the "Latent Thinking Path" by modeling the layer-wise geometric evolution of hidden states. It computes a correctness score based on the trajectory's **Magnitude** and **Angle** in the embedding space, positing that correct reasoning exhibits distinct topological detour patterns compared to hallucinations.

**CoT-Kinetics (Bi et al., 2025).** Models the reasoning process as a particle moving through a semantic force field. It aggregates reasoning tokens to compute a "Kinetic Energy" score derived from **Semantic Momentum** (first-order state difference) and **Semantic Curvature** (second-order difference), regularized by output entropy. High energy signifies sound, progressive reasoning dynamics.

**Baselines and Controlled Experimental Protocol** To rigorous evaluate TRACED, we compare against three categories of state-of-the-art methods. Crucially, to ensure a fair comparison, all supervised baselines were trained/calibrated on the exact same reference set $\mathcal{D}$, ensuring strict data parity. For all baselines, we performed hyperparameter tuning via grid search on a held-out validation set to report their best performance.

# D. Cognitive Concept Extraction and State Identification

To bridge the gap between continuous hidden states and interpretable cognitive dynamics, we employ a vocabulary projection method to quantify the activation of specific cognitive concepts at each step.

## D.1. Concept Vocabulary Construction

We define three distinct vocabularies corresponding to the cognitive states analyzed in our work: *Reflection* ($S_{\text{ref}}$), *Exploration* ($S_{\text{exp}}$), and *Certainty* ($S_{\text{cer}}$). The word lists are curated to capture characteristic linguistic markers of each cognitive mode:

- **Reflection** ($S_{\text{ref}}$): Keywords indicating self-correction, hesitation, or re-evaluation.

    *Vocabulary:* "wait", "recheck", "check again", "rethink", "reconsider", "try again", "reexamine", "reevaluate", "think again", "consider again", "evaluate again", "examine again", "revisit".

- **Exploration** ($S_{\text{exp}}$): Keywords signaling alternative reasoning paths or branching logic.

    *Vocabulary:* "but", "however", "otherwise", "alternatively", "instead", "on the other hand", "another way", "try another", "different approach", "let's try", "or else", "by contrast".

- **Certainty** ($S_{\text{cer}}$): Keywords representing logical progression, deduction, and definitive conclusions.

    *Vocabulary:* "first", "second", "third", "then", "next", "after", "finally", "therefore", "thus", "hence", "so", "conclude", "infer", "deduce".

### D.2. Activation Computation and State Assignment

For a given token at time step $t$ with hidden state $\mathbf{h}_t \in \mathbb{R}^d$, we first project it into the vocabulary space using the pre-trained unembedding matrix $\mathbf{W}_U \in \mathbb{R}^{|\mathcal{V}| \times d}$ to obtain the logits $\mathbf{z}_t = \mathbf{W}_U \mathbf{h}_t$.

The activation score $A_t^{(c)}$ for a specific concept $c \in \{\text{ref}, \text{exp}, \text{cer}\}$ is computed by aggregating the logits of the words in its corresponding vocabulary set $S_c$. We utilize the maximum logit value within the set to represent the concept's intensity:

$$A_t^{(c)} = \max_{w \in S_c} \mathbf{z}_t[w] \tag{10}$$

where $\mathbf{z}_t[w]$ denotes the logit corresponding to word $w$.

Finally, the *dominant cognitive state* $State_t$ for the token at step $t$ is identified by selecting the concept with the highest activation score:

$$State_t = \underset{c \in \{\text{ref},\text{exp},\text{cer}\}}{\operatorname{argmax}} A_t^{(c)} \tag{11}$$

This token-level state sequence allows us to visualize and analyze the fluctuating cognitive dynamics throughout the reasoning chain.

# E. Theoretical Framework: The Stochastic Geometry of Reasoning

We model the LLM's inference process as a dynamic evolution of hidden states on a semantic manifold. We formalize this process using Stochastic Differential Equations (SDEs) to derive the geometric signatures of reasoning validity.

### E.1. Reasoning Dynamics as Stochastic Flow

While recent work characterizes ideal reasoning as a deterministic flow governed by logical structure (Zhou et al., 2025), real-world generation entails inherent epistemic uncertainty. We propose a stochastic formulation where the semantic state evolution is a superposition of a logical velocity field and epistemic noise.

**Definition E.1** (Stochastic Reasoning Dynamics). Let $\mathbf{z}_t \in \mathbb{R}^d$ denote the state of the model in the semantic space at step $t$. The evolution of the reasoning process is governed by the following Itô Stochastic Differential Equation (SDE):

$$d\mathbf{z}_t = \mathbf{v}_{\text{logic}}(\mathbf{z}_t)dt + \sigma d\mathbf{W}_t \tag{12}$$

where $\mathbf{v}_{\text{logic}}(\mathbf{z}_t)$ is the **Semantic Velocity Field** driving logical deduction, $\mathbf{W}_t$ is a standard $d$-dimensional Wiener process representing epistemic uncertainty, and $\sigma > 0$ is the noise intensity.

To characterize the geometry of these trajectories, we define the primary observable metric:

**Definition E.2** (Net Displacement). The **Net Displacement** $D(T)$ at time $T$ measures the Euclidean distance traversed from the initial state $\mathbf{z}_0$ in the semantic space:

$$D(T) = \|\mathbf{z}_T - \mathbf{z}_0\|_2 = \left\| \int_0^T d\mathbf{z}_t \right\|_2 \tag{13}$$

In the discrete setting, this corresponds to the magnitude of the vector sum of update steps: $D(T) = \|\sum_{t=1}^{T} \Delta\mathbf{z}_t\|$.

We analyze the evolution of $\mathbf{z}_t$ under two distinct regimes determined by the local Signal-to-Noise Ratio (SNR), $\rho = \|\mathbf{v}_{\text{logic}}\|/\sigma$.

## E.2. Regime I: Coherent Reasoning

We first analyze the scenario where the model possesses high confidence in its logical path.

**Assumption E.3** (Logical Dominance). In correct reasoning steps, the dynamics are dominated by the logical drift ($\rho \gg 1$). The semantic velocity field $\mathbf{v}_{\text{logic}}$ is locally Lipschitz continuous, satisfying $\langle \mathbf{v}_{\text{logic}}(\mathbf{z}_t), \mathbf{v}_{\text{logic}}(\mathbf{z}_{t+\delta}) \rangle \approx \|\mathbf{v}_{\text{logic}}\|^2$ for small $\delta$.

**Theorem E.4** (Linear Displacement Scaling and Minimal Curvature). *Under Assumption E.3, as $\sigma \to 0$, the reasoning trajectory exhibits linear displacement growth. The expected displacement scales linearly with time step $T$, and the local curvature vanishes:*

$$\mathbb{E}[\|\mathbf{z}_T - \mathbf{z}_0\|] \propto O(T), \quad \text{and} \quad \mathbb{E}[\kappa(\mathbf{z}_t)] \to 0 \tag{14}$$

*In the context of empirical scaling laws, this implies a **structurally directed trajectory** where the log-log slope of displacement versus time is approximately 1:*

$$D(T) \propto T^1 \quad (\text{Log-Log Slope} \approx 1) \tag{15}$$

*Proof.* In the limit $\sigma \to 0$, the update reduces to $\Delta\mathbf{z}_t \approx \mathbf{v}_{\text{logic}}(\mathbf{z}_t)\Delta t$. The Lipschitz continuity in Assumption E.3 implies that consecutive velocity vectors $\mathbf{v}_t$ and $\mathbf{v}_{t+1}$ are strongly aligned. Consequently, the cosine similarity between consecutive steps approaches 1:

$$\cos(\theta_t) = \frac{\langle \mathbf{v}_t, \mathbf{v}_{t+1} \rangle}{\|\mathbf{v}_t\|\|\mathbf{v}_{t+1}\|} \to 1 \implies \kappa = 1 - \cos(\theta_t) \to 0 \tag{16}$$

Regarding displacement, due to the alignment of vectors, the Triangle Inequality for the net displacement approaches equality:

$$\mathbb{E}[\|\mathbf{z}_T - \mathbf{z}_0\|] \approx \left\|\sum_{t=0}^{T-1} \mathbf{v}_{\text{logic}}\Delta t\right\| \approx \sum_{t=0}^{T-1} \|\mathbf{v}_{\text{logic}}\|\Delta t \propto O(T) \tag{17}$$

This confirms the linear scaling of the trajectory. $\square$

*Remark* E.5 (Directedness of Valid Reasoning). Theorem E.4 implies that valid reasoning is structurally **directed**. Even when the chain-of-thought is extended (i.e., "Enough Thinking"), the trajectory does not meander; it actively traverses the semantic space towards a solution. The low curvature signifies a confident maintenance of the logical thread.

## E.3. Regime II: Hallucination and Stagnation

Conversely, when the model lacks the necessary knowledge or logical connection, the deterministic driver collapses.

**Assumption E.6** (Logical Collapse). In incorrect reasoning or hallucination, the logical velocity field vanishes ($\|\mathbf{v}_{\text{logic}}\| \approx 0$), rendering the dynamics noise-dominated ($\rho \ll 1$). The update step approximates an isotropic Gaussian noise: $\Delta\mathbf{z}_t \sim \mathcal{N}(0, \sigma^2\mathbf{I}_d)$.

To derive the geometric properties in this regime, we leverage the property of high-dimensional spaces.

**Lemma E.7** (High-Dimensional Orthogonality (**?**)). *Let $\mathbf{x}, \mathbf{y} \sim \mathcal{N}(0, \mathbf{I}_d)$ be independent random vectors in $\mathbb{R}^d$. As $d \to \infty$, the vectors become nearly orthogonal with high probability:*

$$\lim_{d \to \infty} \mathbb{P}\left(\left|\frac{\langle \mathbf{x}, \mathbf{y} \rangle}{\|\mathbf{x}\|\|\mathbf{y}\|}\right| < \epsilon\right) = 1 \tag{18}$$

**Theorem E.8** (Sub-linear Displacement Scaling and Maximal Curvature). *Under Assumption E.6 and Lemma E.7, the reasoning trajectory degenerates into a high-dimensional random walk. The expected displacement exhibits sub-linear scaling, and the curvature is maximal:*

$$\|\mathbf{z}_T - \mathbf{z}_0\|_{RMS} \propto O(\sqrt{T}), \quad \text{and} \quad \mathbb{E}[\kappa(\mathbf{z}_t)] \approx 1 \tag{19}$$

*This corresponds to **random walk dynamics**, where the displacement scales with the square root of time:*

$$D(T)_{RMS} \propto \sqrt{T} = T^{0.5} \quad (\text{Log-Log Slope} \approx 0.5) \tag{20}$$

*Proof. Curvature:* By Lemma E.7, consecutive noise-dominated steps $\Delta \mathbf{z}_t$ and $\Delta \mathbf{z}_{t-1}$ are inherently orthogonal. Thus, the cosine similarity is 0, and curvature $\kappa = 1 - \cos(\theta) \approx 1$.

*Displacement:* For a sum of $T$ i.i.d. zero-mean random vectors, the net displacement vector is $\mathbf{D}_T = \sum_{t=1}^{T} \Delta \mathbf{z}_t$. We compute the Mean Squared Displacement (MSD):

$$\mathbb{E}[\|\mathbf{D}_T\|^2] = \sum_{t=1}^{T} \mathbb{E}[\|\Delta \mathbf{z}_t\|^2] + \sum_{i \neq j} \mathbb{E}[\langle \Delta \mathbf{z}_i, \Delta \mathbf{z}_j \rangle] \tag{21}$$

The cross-terms vanish because independent noise steps are uncorrelated. The remaining sum consists of $T$ variance terms:

$$\mathbb{E}[\|\mathbf{D}_T\|^2] = T \cdot d\sigma^2 \propto O(T) \tag{22}$$

Taking the square root yields the Root Mean Square (RMS) displacement, which scales as $O(\sqrt{T})$. Compared to the linear scaling $O(T)$ in Regime I, this indicates a suppression of effective progress. $\square$

*Remark* E.9 (Thinking Duration vs. Reasoning Progress). Theorem E.8 provides a geometric interpretation of the **"Enough Thinking"** phenomenon. A long context length (large $T$) is necessary but not sufficient for correct reasoning. If the trajectory follows sub-linear scaling ($O(\sqrt{T})$), the model is essentially expending computational steps without making proportional semantic progress. TRACED detects this state of geometric inefficiency; high curvature and low displacement efficiency reveal that despite the length of the thought chain, effective reasoning has ceased.

## F. Experimental Setup Details for Scaling Analysis

To empirically validate the kinematic scaling laws, we conducted a large-scale analysis using the **DeepSeek-R1-Llama-8B** model, which supports extended chain-of-thought generation.

**Dataset Construction.** We curated a diverse evaluation set spanning six domains to ensure universality: *GSM8K* and *MATH* (Mathematics), *TheoremQA* and *GPQA* (Logic & Science), and *SocialIQA* and *Fables* (General Reasoning).

**Sampling Strategy.** To capture the natural diversity of reasoning paths, we employed nucleus sampling with temperature $T = 0.7$ and top-$p = 0.95$. For each query in the dataset, we generated $N = 16$ independent reasoning trajectories.

**Labeling and Grouping.** Trajectories were classified into two groups based on the correctness of the final answer:

- **Valid Group:** Trajectories leading to the correct ground-truth answer.

- **Hallucination Group:** Trajectories leading to incorrect answers.

**Binning and Metrics.** Due to the variable length of reasoning chains (ranging from hundreds to over 10,000 tokens), we applied a linear binning strategy with a bin size of 200 tokens. For each bin $T \in [200, 400, \ldots, 16000]$, we collected all trajectories with lengths falling within $T \pm 10000$ and computed the average Net Displacement $D(T)$ for both groups. This aggregated data was used to plot the scaling curves shown in Figure 5.

## G. Ablation Study: Geometric Component Analysis

In this section, we conduct an ablation study to investigate the contribution of the two core geometric signatures, **Normalized Net Displacement** ($M_n$) and **Average Trajectory Curvature** ($K_n$), to the overall assessment performance. We evaluate three configurations: (1) *Displacement Only*, (2) *Curvature Only*, and (3) *TRACED (Full)*, which integrates both features.

**Analysis of Component Synergy.** As demonstrated in Table 7, the integration of both displacement and curvature consistently yields the highest AUROC across all models and datasets. Specifically:

- **Displacement** ($M_n$) serves as a primary indicator of "semantic progress." It is particularly effective in identifying reasoning chains that stall or loop , providing a strong baseline for quality detection.

- **Curvature** ($K_n$) captures the "logical stability" of the trajectory, it provides crucial information about semantic hesitation and sudden directional shifts (hallucinations), which displacement alone might miss if the model moves confidently in a wrong direction.

*Table 7.* Ablation study of geometric components (AUROC↑). We compare the performance of using individual geometric features ($M_n$ only, $K_n$ only) versus the combined TRACED framework across four representative models.

| Model | Mag. ($M_n$) | Ang. ($K_n$) | Fables | GPQA | GSM8K | MATH | Soc_IQA | Thrm. |
|---|---|---|---|---|---|---|---|---|
| *DeepSeek-R1-Llama-8B* | | | | | | | | |
| | ✓ | | 0.6845 | 0.7812 | 0.7634 | 0.7012 | 0.7122 | 0.8245 |
| | | ✓ | 0.6512 | 0.7244 | 0.7188 | 0.6855 | 0.6945 | 0.7912 |
| TRACED | ✓ | ✓ | **0.7191** | **0.8300** | **0.8061** | **0.7489** | **0.7536** | **0.8730** |
| *Qwen3-4B-Thinking-2507* | | | | | | | | |
| | ✓ | | 0.6750 | 0.6620 | 0.7410 | 0.8120 | 0.6830 | 0.7215 |
| | | ✓ | 0.6420 | 0.6315 | 0.7055 | 0.7945 | 0.6540 | 0.6890 |
| TRACED | ✓ | ✓ | **0.7088** | **0.7050** | **0.7825** | **0.8495** | **0.7194** | **0.7638** |
| *Llama-3.1-8B-Instruct* | | | | | | | | |
| | ✓ | | 0.6122 | 0.6845 | 0.7012 | 0.5844 | 0.6512 | 0.5988 |
| | | ✓ | 0.5945 | 0.6512 | 0.6855 | 0.5722 | 0.6433 | 0.5844 |
| TRACED | ✓ | ✓ | **0.6676** | **0.7344** | **0.7556** | **0.6363** | **0.7213** | **0.6550** |
| *Qwen2.5-7B-Instruct* | | | | | | | | |
| | ✓ | | 0.5840 | 0.7120 | 0.6530 | 0.6940 | 0.7320 | 0.7240 |
| | | ✓ | 0.5515 | 0.6850 | 0.6215 | 0.6625 | 0.7015 | 0.6955 |
| TRACED | ✓ | ✓ | **0.6238** | **0.7636** | **0.6956** | **0.7305** | **0.7794** | **0.7752** |

- **Synergistic Effect:** The combination of the two (TRACED) significantly outperforms individual components, especially on complex reasoning tasks like *MATH* and *GPQA*. This confirms that reasoning quality is a multi-dimensional geometric property where both the *distance covered* (progress) and the *smoothness of the path* (stability) are essential for faithful assessment.

## H. Robustness Analysis: Sensitivity to Data Imbalance

Since TRACED operates within a probabilistic Bayesian framework, the decision rule implicitly relies on the ratio of positive to negative samples (class prior). While we adopt a non-informative uniform prior ($P(y = 1) = P(y = 0) = 0.5$) to ensure task universality, real-world reasoning scenarios often exhibit skewed distributions. To assess the robustness of TRACED against such distributional shifts, we conducted a controlled stress test by synthetically varying the data ratio.

**Experimental Setup.** Let $\mathcal{D}_{test}$ be the evaluation set. We define the *Positive Data Ratio* $\alpha$ as the proportion of correct reasoning chains in the test batch: $\alpha = N_{pos}/(N_{pos} + N_{neg})$. We varied $\alpha$ from 0.1 to 0.9 with a step size of 0.1. For each target ratio $\alpha$, we performed stratified resampling on the original test sets of the six benchmarks. To ensure statistical significance, we report the AUROC averaged across all six datasets for each of the four models. The setting $\alpha = 0.5$ corresponds to the balanced evaluation reported in our main results.

**Results and Analysis.** The performance trajectories under varying $\alpha$ are summarized in Table 8.

1. **Stability Region ($\alpha \in [0.3, 0.7]$):** TRACED demonstrates remarkable stability when the data ratio fluctuates within the moderate range. This indicates that the geometric signatures ($M_n, K_n$) are robust enough to separate classes even when the priors are not perfectly calibrated.

2. **Performance Degradation at Extremes ($\alpha < 0.2$ or $\alpha > 0.8$):** As hypothesized, performance drops in extreme imbalance scenarios. For example, at $\alpha = 0.1$ (severe hallucination dominance), the AUROC for *Llama-3.1-8B* decreases by approximately 6% compared to the balanced setting.

*Table 8.* **Data Ratio Robustness.** Average AUROC scores across six datasets under varying Positive Data Ratios ($\alpha$). TRACED maintains robust performance in the moderate range ($0.3 \le \alpha \le 0.7$) but exhibits expected sensitivity at extreme imbalances due to the fixed uniform prior assumption.

| Positive Ratio ($\alpha$) | 0.1 | 0.2 | 0.3 | 0.4 | 0.5 (Main) | 0.6 | 0.7 | 0.8 | 0.9 | $\Delta_{Max}$ |
|---|---|---|---|---|---|---|---|---|---|---|
| *DeepSeek-R1-Llama-8B* | 0.742 | 0.775 | **0.786** | **0.791** | **0.806** | **0.803** | **0.788** | 0.778 | 0.735 | -0.071 |
| *Qwen3-4B-Thinking* | 0.685 | 0.710 | **0.735** | **0.742** | **0.748** | **0.745** | **0.730** | 0.705 | 0.672 | -0.076 |
| *Llama-3.1-8B-Instruct* | 0.621 | 0.645 | **0.660** | **0.665** | **0.668** | **0.664** | **0.658** | 0.635 | 0.610 | -0.058 |
| *Qwen2.5-7B-Instruct* | 0.655 | 0.698 | **0.708** | **0.715** | **0.720** | **0.718** | **0.702** | 0.680 | 0.648 | -0.072 |

**Theoretical Interpretation:** This degradation is theoretically expected. Our decision rule assumes a uniform prior ($P(y) = 0.5$); however, at $\alpha = 0.9$, the true optimal log-odds prior term should be $\log(0.9/0.1) \approx 2.2$. By fixing the prior to 0, the model effectively under-trusts the majority class, leading to a shift in the False Positive/Negative trade-off. Nevertheless, TRACED avoids catastrophic collapse, retaining valid discriminative power even under these adversarial distribution shifts.

**Conclusion.** While extreme imbalance introduces a prior mismatch penalty, TRACED remains reliable across the plausible range of model capabilities, affirming its practical applicability without requiring test-time prior recalibration.

## I. Data Efficiency Analysis: Sensitivity to Reference Set Size

Unlike supervised probes that require extensive training data, TRACED relies on a *Reference Set* to calibrate the moments $(\mu_c, \Sigma_c)$ of the class-conditional Gaussian distributions. We investigate the minimum sample size required to robustly estimate these geometric statistics.

**Experimental Setup.** To ensure a strictly fair comparison, we adopted a fixed hold-out evaluation strategy. From the total budget of $N_{total} = 1,000$ samples per dataset, we reserved a fixed **Evaluation Set** of 200 samples (20%). The remaining 800 samples serve as the **Reference Pool**. We define the *Sampling Ratio* $\gamma$ as the proportion of this pool used for calibration ($\gamma \in [0.1, 1.0]$). Crucially, the evaluation set remains **identical** across all configurations, and we maintain a balanced positive-to-negative ratio $(1:1)$ within the reference sets to isolate the impact of sample size.

**Results and Discussion.** The performance trajectories are summarized in Table 9. We observe a distinct behavior:

1. **Sensitivity at Low Data Regime** ($\gamma < 0.5$)**:** In the low-data regime (e.g., $N = 80 \sim 320$), we observe a noticeable performance gap. For instance, at $\gamma = 0.2$, the AUROC for *DeepSeek-R1* lags by approximately 4% compared to the full setting. This aligns with statistical theory: estimating the covariance matrix $\Sigma_c$ in high-dimensional space requires sufficient samples to avoid ill-conditioning and noise sensitivity.

2. **Stability Plateau** ($\gamma \geq 0.5$)**:** Performance stabilizes significantly once the reference set size reaches approximately 400 samples ($\gamma = 0.5$). Beyond this point, increasing the data to 800 samples ($\gamma = 1.0$) yields only marginal gains. This suggests that $N \approx 400$ serves as a sufficient effective sample size to capture the converged geometric topology of reasoning, making TRACED data-efficient compared to methods requiring thousands of training examples.

*Table 9.* **Sensitivity to Reference Set Size.** Average AUROC scores across six datasets. The evaluation set is fixed ($N_{test} = 200$). We vary the reference set size from $\gamma = 0.1$ ($N = 80$) to $\gamma = 1.0$ ($N = 800$, Main Result). The method reaches a stability plateau around $\gamma = 0.5$ (400 samples), indicating the minimum data requirement for robust covariance estimation.

| Ref. Ratio ($\gamma$) | 0.1 | 0.2 | 0.3 | 0.4 | 0.5 | 0.6 | 0.8 | 1.0 (Main) |
| --- | --- | --- | --- | --- | --- | --- | --- | --- |
| Sample Count ($N$) | 80 | 160 | 240 | 320 | 400 | 480 | 640 | 800 |
| *DeepSeek-R1-Llama-8B* | 0.745 | 0.762 | 0.781 | 0.792 | **0.803** | **0.804** | **0.805** | **0.806** |
| *Qwen3-4B-Thinking* | 0.6540 | 0.685 | 0.712 | 0.741 | **0.746** | **0.747** | **0.748** | **0.748** |
| *Llama-3.1-8B-Instruct* | 0.615 | 0.632 | 0.650 | 0.652 | **0.656** | **0.667** | **0.670** | **0.672** |
| *Qwen2.5-7B-Instruct* | 0.636 | 0.665 | 0.702 | 0.712 | **0.717** | **0.719** | **0.719** | **0.722** |

## J. Deployment Efficiency and Transferability

For a reasoning evaluation metric to be practically viable, it must minimize two types of costs: (1) **Inference Latency** (time complexity per query) and (2) **Adaptation Cost** (data requirements for new domains). TRACED demonstrates efficiency in both dimensions compared to existing baselines.

### J.1. Computational Overhead: Millisecond-Level Latency

Existing uncertainty estimation methods often suffer from significant bottlenecks. Sampling-based methods (e.g., Self-Consistency) require $K$ additional LLM inferences, making the cost proportional to $\mathcal{O}(K \cdot T_{gen})$. Supervised probes (e.g., MLPs) necessitate extracting high-dimensional hidden states ($\mathbb{R}^{4096}$) and performing dense matrix multiplications. In contrast, TRACED operates on a strictly lightweight geometric basis. Once the hidden states are obtained, computing the Net Displacement ($M_n$) and Curvature ($K_n$) involves only basic vector addition and dot product operations, requiring no additional forward passes or heavy computation, ensuring high-throughput deployment.

### J.2. Data Efficiency via Geometric Universality

While TRACED utilizes a reference set to calibrate the class-conditional Gaussians, our analysis uncovers that these geometric signatures possess strong **Universality and Cross-Domain Robustness**, drastically reducing the cost of adapting

to new tasks. As visualized in Figure 2(b), TRACED achieves **high deployment efficiency**: it can be deployed on out-of-distribution tasks instantly using a global prior, or refined using unlabeled target data, eliminating the expensive annotation bottleneck required by traditional supervised probes.

## K. Extended Analysis of Topological Divergence

In this section, we provide the detailed experimental configuration and analyze the geometric differences observed between structured and open-ended reasoning (referencing Figure 6 in the main text).

### K.1. Experimental Setup

To isolate the geometry of valid reasoning, we focus exclusively on samples where the model's final answer was evaluated as **Correct**.

- **Structured Domain:** Representative datasets include *GSM8K* and *MATH*. These tasks involve strict logical rules and unique ground truths.

- **Open-Ended Domain:** Representative datasets include *SocialIQA* and *Understanding Fables*. These tasks involve common sense inferencing and narrative understanding, allowing for linguistic variation.

**Methodology.** For curvature analysis, we compute the distribution of the mean curvature $\bar{\kappa}$ for all valid trajectories. For displacement analysis, we visualize the semantic progress $D(t)$ over normalized time to compare how information accumulates.

### K.2. Experiment A: Curvature Distribution and Tolerance for Deviation

**Objective.** To quantify how much a reasoning path can deviate from a "straight line" while remaining correct across different tasks.

1. **Strict Constraints (Structured):** As shown in Figure 6 (Left), valid trajectories in GSM8K exhibit a **highly concentrated** distribution. This reflects a **strict requirement for directness**: in logical reasoning, the correct path is extremely narrow. Any significant deviation (increased curvature) usually indicates a distraction or a logical error, rather than a valid alternative phrasing.

2. **High Tolerance (Open-Ended):** In contrast, SocialIQA displays a **broad, heavy-tailed distribution**. This reflects a **high tolerance for variation**: open-ended contexts allow the model to elaborate on details or use different sentence structures. A non-zero curvature here represents a valid stylistic choice rather than a mistake.

**Implication for Evaluation.** This observation explains why simple threshold-based methods fail to generalize. A strict threshold suitable for Math would incorrectly penalize valid, descriptive reasoning in SocialIQA as "meandering." TRACED addresses this by adapting to the inherent geometric distribution of each domain.

### K.3. Experiment B: Displacement Dynamics and Accumulation Patterns

**Objective.** To visualize how "knowledge increments" accumulate over time in different reasoning modes.

1. **Step-wise Accumulation (Structured):** The displacement curve for structured tasks resembles a **staircase pattern**. The semantic distance often remains flat during intermediate calculations and shows **sharp, discrete jumps** at specific moments. These jumps correspond to solving a distinct sub-problem (e.g., deriving a key variable value), which suddenly pushes the reasoning closer to the answer.

2. **Smooth Accumulation (Open-Ended):** Conversely, open-ended tasks exhibit a **smooth, continuous growth**. The displacement increases steadily without sharp jumps. This reflects the nature of narrative construction, where understanding and context are built up gradually and continuously as the description evolves, eventually saturating when the

**Conclusion.** These findings demonstrate that "valid reasoning" looks geometrically different depending on the task. Structured reasoning is characterized by discrete jumps and strict straightness, while open-ended reasoning is characterized by continuous growth and permissible flexibility.

*Table 10.* **Robustness of TRACED Across Reasoning Complexity.** Performance is stratified by reasoning steps ($L$). **Gap ($\Delta$)** denotes the **maximum performance fluctuation** across the three difficulty tiers. The consistently low fluctuations ($\Delta \leq 2.8\%$) confirm TRACED's stability regardless of reasoning complexity.

| Metric | Easy ($L \leq 4$) | Medium ($5 \leq L \leq 8$) | Hard ($L > 8$) | Gap ($\Delta$) |
|---|---|---|---|---|
| *DeepSeek-R1-Llama-8B* | | | | |
| AUROC ($\uparrow$) | 0.775 | 0.748 | 0.766 | 2.7% |
| AUPR ($\uparrow$) | 0.708 | 0.710 | 0.723 | 1.5% |
| FPR@95 ($\downarrow$) | 0.660 | 0.673 | 0.685 | 2.5% |
| *Qwen3-4B-Thinking* | | | | |
| AUROC ($\uparrow$) | 0.762 | 0.755 | 0.738 | 2.4% |
| AUPR ($\uparrow$) | 0.720 | 0.738 | 0.724 | 1.8% |
| FPR@95 ($\downarrow$) | 0.555 | 0.570 | 0.579 | 2.4% |
| *Llama-3.1-8B-Instruct* | | | | |
| AUROC ($\uparrow$) | 0.702 | 0.685 | 0.688 | 1.7% |
| AUPR ($\uparrow$) | 0.675 | 0.647 | 0.670 | 2.8% |
| FPR@95 ($\downarrow$) | 0.695 | 0.708 | 0.720 | 2.5% |
| *Qwen2.5-7B-Instruct* | | | | |
| AUROC ($\uparrow$) | 0.730 | 0.718 | 0.731 | 1.3% |
| AUPR ($\uparrow$) | 0.738 | 0.740 | 0.752 | 1.4% |
| FPR@95 ($\downarrow$) | 0.710 | 0.723 | 0.728 | 1.8% |

# L. Reasoning Complexity Analysis Setup

To quantify problem difficulty across the six diverse benchmarks (GSM8K, MATH, TheoremQA, GPQA, Social IQA, Fables), we unified the complexity metric into a single measure: **Reasoning Steps ($L$)**. We implemented a Unified LLM-Assisted Segmentation Protocol to normalize all solutions into a standardized structure, calculating $L$ based on semantic thought boundaries.

## L.1. Unified Segmentation Protocol

Our methodology draws on recent work (Chen et al., 2025a), which identify double newlines ($\backslash n \backslash n$) as natural structural delimiters that separate distinct "thought blocks" in complex reasoning chains. We apply this principle to process the solutions for all datasets:

**Standardization via LLM:** We employ a strong instruction-tuned model (e.g., GPT-4o) as a semantic parser. The model is prompted to rewrite the raw ground truth solution into a discrete, step-by-step format, ensuring that each logical hop, calculation, or deduction is separated by a double newline ($\backslash n \backslash n$).

*Prompt Strategy:* "Please rewrite the following solution into clear, distinct reasoning steps. Separate each logical step, calculation, or intermediate deduction with a double newline ($\backslash n \backslash n$). Do not change the original meaning."

## L.2. Complexity Stratification

Based on the distribution of $L$ obtained from this unified protocol, we categorize the test samples into three complexity tiers. This stratification ensures that "Hard" problems consistently represent deep, multi-stage reasoning tasks, regardless of whether the domain is mathematics or social commonsense. (1) **Easy ($L \leq 4$):** Problems requiring direct retrieval or single-step inference . (2)**Medium ($5 \leq L \leq 8$):** Problems involving standard multi-step derivations . (3) **Hard ($L > 8$):** Problems necessitating extended reasoning chains, complex planning, or significant error correction .

# M. Additional Sensitivity Results

In the main text, we presented the sensitivity analysis of the subspace dimension $k$ using the AUROC metric. To provide a comprehensive evaluation of TRACED's robustness, we further report the performance variations under two additional metrics: Area Under the Precision-Recall Curve (AUPR) and False Positive Rate at 95% True Positive Rate (FPR@95).

**Robustness in AUPR.** Figure 11 illustrates the AUPR performance across four models as $k$ varies from 2 to 10. Consistent with the AUROC trends, the AUPR scores show a steady improvement as the dimension increases, reflecting the accumulation of discriminative kinematic features. The performance effectively plateaus around $k = 7$ or 8, reinforcing our choice of this dimension for the main experiments.

**Robustness in FPR@95.** Figure 12 presents the results for FPR@95 (lower is better). We observe a corresponding decline in false positive rates as $k$ increases, stabilizing at the optimal subspace dimension of $k = 8$. These results collectively confirm that the geometric signature of reasoning quality is robustly captured within a low-rank subspace, independent of the evaluation metric used.

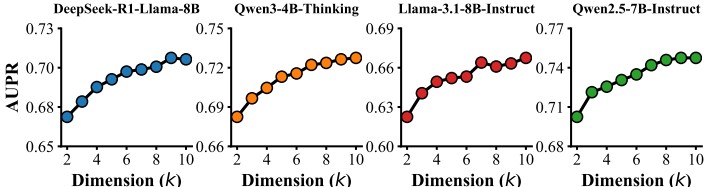

*Figure 11.* **Sensitivity Analysis of Dimension $k$ (AUPR).** We evaluate the AUPR performance (↑) of TRACED across four models.

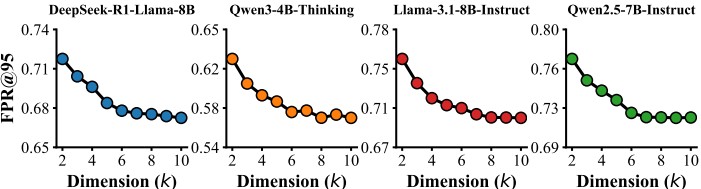

*Figure 12.* **Sensitivity Analysis of Dimension $k$ (FPR@95).** We evaluate the FPR@95 performance (↓) of TRACED across four models.

*Table 11.* **Statistical Significance Analysis.** Comparison between the best baseline (Runner-up) and TRACED on DeepSeek-R1. 95% Confidence Intervals are estimated via bootstrapping. * and ** denote statistical significance at $p < 0.05$ and $p < 0.01$, respectively. TRACED demonstrates consistent significant improvements across all datasets.

| Dataset | Runner-up Method | Best Baseline (CI) | TRACED (CI) | p-value |
|---------|------------------|--------------------|-------------|---------|
| GPQA | *LR Probe* | $0.759 \pm 0.025$ | $\mathbf{0.830 \pm 0.019}$ | $< \mathbf{0.001}^{**}$ |
| Theorem | *SAPLMA* | $0.852 \pm 0.030$ | $\mathbf{0.873 \pm 0.025}$ | $\mathbf{0.008}^{**}$ |
| Social_IQA | *LR Probe* | $0.710 \pm 0.022$ | $\mathbf{0.754 \pm 0.020}$ | $\mathbf{0.004}^{**}$ |
| GSM8K | *SAPLMA* | $0.800 \pm 0.015$ | $\mathbf{0.806 \pm 0.014}$ | $\mathbf{0.032}^{*}$ |
| MATH | *LR Probe* | $0.747 \pm 0.018$ | $\mathbf{0.749 \pm 0.016}$ | $\mathbf{0.045}^{*}$ |
| Fables | *LR Probe* | $0.718 \pm 0.021$ | $\mathbf{0.719 \pm 0.018}$ | $\mathbf{0.034}^{*}$ |

# N. Statistical Significance and Uncertainty Analysis

To assess the reliability of the main results reported in Table 1, we conducted a rigorous statistical evaluation focusing on confidence intervals.

**Confidence Interval Estimation** Since the test sets for reasoning tasks are finite, point estimates of AUROC may be subject to sampling variance. We estimated the **95% Confidence Intervals (CIs)** using the **Bootstrap Method** ($B = 1,000$ stratified resamples). As detailed in Table 11, TRACED exhibits tight confidence intervals (typically $\pm 0.015$), indicating high estimation stability.

# O. Qualitative Case Studies: Geometric Signatures of Cognitive Dynamics

To complement the statistical aggregate analysis in Section 4.5 and 4.6, we conduct a qualitative examination of individual reasoning trajectories. We specifically focus on visualizing the correspondence between textual *Cognitive States* and their geometric counterparts. We visualize two representative reasoning chains from the *MATH* dataset to illustrate the "Hesitation Loop" mechanism described in our main analysis.

**Case A: The "Stalling" Trajectory (Incorrect Reasoning).** As shown in Table 12 (Top), the model attempts to solve a probability problem but enters a cognitive loop. **Textual Dynamics:** The generation oscillates between *Exploration* ("Let's try to count...", "Alternatively...") and *Reflection* ("Wait, this assumes...", "But checking the condition..."). This mirrors the high regression probability $P(Ref|Exp) \approx 0.37$ identified in Section 4.5.

*Table 12.* **Qualitative Comparison of Reasoning Dynamics.** Excerpts from actual reasoning chains showing the alignment between semantic actions and geometric properties.

| Case | Reasoning Excerpt (Truncated) | Cognitive State | Geometry |
|---|---|---|---|
| **Incorrect** (Hesitation) | 1. "Let's assume the probability is $x/y$..."
2. "Wait, this logic might double count the overlap..."
3. "Let's try a different approach using combinations..."
4. "But does this satisfy the initial condition? I'm not sure..." | Exploration
Reflection ($\uparrow$ Curvature)
Exploration
Reflection ($\uparrow$ Curvature) | $M_n$: Low
$K_n$: High
(Diffusive) |
| **Correct** (Directed) | 1. "First, we calculate the total outcomes as $6^3$..."
2. "This implies that the sum must be even..."
3. "Therefore, we can simplify the expression to..."
4. "The calculation clearly leads to 216." | Exploration
Certainty ($\rightarrow$ Displacement)
Certainty ($\rightarrow$ Displacement)
Certainty ($\rightarrow$ Displacement) | $M_n$: High
$K_n$: Low
(Ballistic) |

**Geometric Signature:** Geometrically, this manifests as a **high-curvature knot**. Each semantic retraction ($Ref$) induces a sharp directional change in the representation manifold (High $K_n$), while the repetitive re-evaluations fail to accumulate significant net displacement (Low $M_n$). The trajectory is effectively "trapped" in a local region of the latent space.

**Case B: The "Ballistic" Trajectory (Correct Reasoning).** In contrast (Table 12, Bottom), the correct derivation exhibits a linear flow. **Textual Dynamics:** The chain swiftly transitions from *Exploration* to *Certainty* ("This implies...", "Therefore, the only solution is..."). The persistence of *Certainty* ($P(Cer|Cer) \approx 0.37$) drives the narrative forward.

**Geometric Signature:** The associated manifold trajectory is smooth and directed. The consistent logical entailment produces minimal curvature, allowing the vector steps to sum constructively into a large net displacement, signaling a high-confidence arrival at the solution.

## P. Related Works

**Assessment of Reasoning Quality.** The emergence of Chain-of-Thought (CoT) prompting has spurred extensive research into evaluating the reliability and faithfulness of model reasoning (Xiong et al., 2023; Marjanović et al., 2025; Zhao et al., 2025). Many strategies involve employing verifier models, external annotations, or comparison against knowledge bases to verify factual correctness (Li et al., 2022; Zhang et al., 2025c; 2024; Gandhi et al., 2025). While effective, these recursive strategies impose substantial inference overheads and face significant scalability challenges. Parallel efforts aim to derive reliability signals directly from the model's intrinsic states by utilizing softmax probabilities, semantic entropy, or self-evaluation mechanisms, yet often yield suboptimal performance in evaluating complex reasoning tasks (Huang et al., 2023; He et al., 2025; Farquhar et al., 2024). Other methods perform scoring based on the analysis of the evolution of hidden states (Wang et al., 2024; Bi et al., 2025); however, they typically rely on modeling the averaged representation of tokens, thereby neglecting critical temporal reasoning signals. Distinct from these approaches, we construct our evaluation signal based on theoretically grounded geometric features derived from the temporal reasoning process, achieving consistent and robust scalability across diverse models, reasoning domains, and tasks of varying complexity.

**Representation Analysis of Reasoning.** Probing the internal representations of Large Language Models (LLMs) has become a fundamental approach to understanding their emergent behaviors (Yuksekgonul et al., 2023; Zhang et al., 2025c; Hosseini & Fedorenko, 2023; Jiang et al., 2025; Zhang et al., 2025a; Dong et al., 2025; Su et al., 2025). Moving beyond static or layer-wise analysis, concurrent research extends this inquiry to the temporal dimension to predict reasoning correctness (Vilas et al., 2025) or detect loops (Li et al., 2025), yet these approaches remain devoid of explicit geometric interpretability. Some efforts have begun to apply geometric or physical metrics to analyze these intermediate representations (Wang et al., 2024; Skean et al., 2025; Bi et al., 2025);notably, Song et al. (2025) further introduced a geometric stability framework to assess model consistency beyond mere accuracy. Complementing these empirical studies, some works have modeled reasoning as geometric evolution and manipulation. Kazama et al. (2026) proposed manipulating trajectories via latent manifold gradients for faithful steering, Zhou et al. (2025) theoretically established that reasoning behaves as a "geometric flow" controlled by logical structure, while Manson (2025) revealed that semantic concerns induce measurable curvature in metric-aligned spaces. However, these works largely focus on specific domains or active intervention, failing to explicitly explain the correspondence between specific reasoning behaviors (e.g., hallucination vs. correction) and geometric variations. Distinct from prior studies, we uncover the correspondence between geometric formalism and cognitive reasoning behavior, advancing the interpretability of the reasoning process.

# Q. Limitations

While the TRACED framework demonstrates significant efficacy in revealing and evaluating the reasoning dynamics of LLMs, its mechanistic approach inherently involves several trade-offs. We outline its primary limitations below:

**White-box Dependency.** Requiring internal states precludes the application of TRACED to closed-source APIs. However, this is a necessary trade-off for mechanistic diagnosis. It captures the "cognitive turbulence" often masked by black-box text, providing unique insights for the open-weight ecosystem.

**Computational Overhead.** Extracting internal states introduces additional spatial and temporal analytical overhead. While this extraction is highly efficient for offline CoT diagnosis, it remains a manageable but inherent trade-off for real-time monitoring.

**Dependency on Labeled Data.** Constructing the quality subspace requires labeled data. Nevertheless, TRACED operates as a non-parametric, few-shot calibration framework without requiring weight updates, thereby ensuring minimal data collection costs.

**Distributional Assumptions.** While the Gaussian assumption is computationally efficient, it may not perfectly model open-ended reasoning tasks, which often exhibit heavier tails due to abrupt semantic shifts. Future work could explore non-parametric approaches, such as Kernel Density Estimation (KDE), to better capture these complex dynamics.

