# OpenReview forum: "Beyond Scalars: Evaluating and Understanding LLM Reasoning via Geometric Progress and Stability"
_ICML.cc/2026/Conference — ICML 2026 regular_

### Official Review · Reviewer_GYLU · 2026-03-08

**Soundness:** 2
**Presentation:** 2
**Significance:** 2
**Originality:** 2
**Overall Recommendation:** 4
**Confidence:** 3

**Summary:**

This paper proposes TRACED, a method for evaluating the quality of LLM reasoning trajectories from hidden-state dynamics. The main idea is to project hidden-state sequences into a learned reasoning-quality subspace and characterize each reasoning trace using two geometric quantities: displacement and curvature. These features are then used in a lightweight Bayesian classifier to distinguish correct from incorrect reasoning trajectories. The paper evaluates the method across multiple benchmarks and model families, and compares it with scalar-confidence methods, hidden-state probes, and prior trajectory-based baselines. In addition to the main performance results, the paper includes analyses on scaling behavior, task transfer, class imbalance, and interpretability. The authors further suggest that different geometric trajectory patterns may correspond to cognitive states such as exploration, hesitation, and certainty.

**Compliance With Llm Reviewing Policy:**

Affirmed.

**Final Justification:**

My concerns are addressed.

**Key Questions For Authors:**

see weaknesses

**Limitations:**

no, see weaknesses

**Strengths And Weaknesses:**

Strength:

1.	The paper introduces a novel geometric view of reasoning evaluation. Representing reasoning as a trajectory and summarizing it with displacement and curvature is intuitive and conceptually appealing.

2.	The experiments cover several benchmarks and different model families, and the method appears competitive with a range of baselines. The paper also includes additional analyses beyond the main results.

3.	Even though not fully convincing yet, the attempt to connect hidden-state trajectory patterns with reasoning behavior is interesting and gives the work broader appeal.

Weaknesses:

1.	The theoretical claims are stronger than the evidence. The paper provides a geometric and stochastic-process interpretation of correct versus incorrect reasoning, but this analysis relies on rather stylized assumptions, especially modeling incorrect reasoning as noise-dominated and approximately isotropic Gaussian. Therefore, the theory reads more as an intuition for why displacement and curvature may be useful, rather than a strong justification that these quantities generally characterize real reasoning dynamics.

2.	A key issue is that the reasoning-quality subspace is explicitly constructed from sets of correct and incorrect trajectories. As a result, the representation itself already depends on labeled supervision, making the method less purely intrinsic than the paper sometimes suggests.

3.	The Bayesian classifier assumes a Gaussian class-conditional structure over the two geometric features, and the paper justifies this mainly via a CLT-style argument. However, the paper provides limited direct validation that this assumption is well matched to the actual feature distributions, making it somewhat unclear how robust the method would be if the Gaussian approximation is poor.

4.	For mathematical reasoning, the paper mainly evaluates on GSM8K and MATH, whereas current mainstream evaluations more commonly include harder benchmarks such as AIME 2024/2025. This makes the empirical support less convincing for modern reasoning models.

---

> ### Author Rebuttal · Authors · 2026-03-29
>
> >  1.Theoretical claims and evidence (W1)
>
> We appreciate the reviewer’s rigorous scrutiny. The SDE-based Gaussian noise model in Appendix E is an idealized abstraction designed to provide **theoretical intuition and motivation** for our metrics, rather than a **rigorous, universal proof** for the complex, non-linear dynamics of real-world LLMs. Recognizing our original phrasing overclaimed mathematical certainty, we revised the text to align theory with empirical evidence:
>
> > Line 165: To operationalize …… reasoning quality under a stochastic differential equation framework.
>
> **Revised to**：To operationalize these physical concepts, we map them to specific geometric features **motivated by** our theoretical analysis. As **formulated** in Appendix E, under an **idealized** stochastic differential equation framework, Displacement and Curvature naturally emerge as **effective geometric proxies** for characterizing reasoning quality.
>
> > 2.Label-Driven Analysis of the Reasoning Subspace (W2)
>
> This aligns with Mnzw; details omitted to save space. See ``2.Label-Driven Analysis of the Quality Subspace`` in response to Mnzw.
>
> > 3.Robustness Analysis of Gaussian Approximation (W3)
>
> To bridge the theoretical Gaussian assumption and empirical long-tailed distributions, our implementation employs **robust estimation**: we filter extreme outliers from the calibration data used for the label-calibrated subspace and Gaussian MAP $P(x|y)$. Fitting these compact Gaussian profiles ($\mathcal{N}\_{pos}, \mathcal{N}\_{neg}$) ensures stable evaluation on the test set (unfiltered ).
>
> Addressing the concern, we compared our Gaussian MAP against flexible models (GMM $n=3$, non-parametric KDE) that capture heavy tails without strict assumptions. Using $N=400$ unfiltered samples, we evaluated structured and open-ended tasks:
>
> |Dataset|Gaussian|GMM|KDE|
> |:---------|:--------|:--------|:--------|
> |MATH|0.627|0.624|**0.628**|
> |GSM8K|0.773|**0.779**|0.773|
> |TheoremQA|0.675|0.673|**0.676**|
> |GPQA|**0.679**|0.668|0.671|
> |SocialIQA|0.633|**0.645**|0.641|
> |Fables|0.629|0.641|**0.644**|
>
> **Structured Tasks:** Features are highly clustered. Flexible models (GMM/KDE) yield marginal gains (<1%) on MATH/GSM8K/Theorem and underperform on GPQA due to overfitting, proving Gaussian MAP's robust competitiveness.
>
> **Open-Ended Tasks:** Semantic divergence (e.g., creative reasoning) introduces long tails. GMM and KDE fit these better, achieving larger gains (Social IQA 1.2% and Fables 1.5%) over Gaussian MAP, which exhibits some sensitivity to unfiltered long tails in these specific areas.
>
> Our core innovation lies in uncovering geometric dynamics, not chasing marginal gains via complex estimators. While KDE may slightly aid heavy-tailed tasks, our Gaussian assumption optimally balances inference efficiency and robustness (analytical $O(1)$ evaluation vs. KDE's $O(N)$ cost). Additionally, filtering outliers stabilizes Gaussian MAP for core samples, driving consistent cross-dataset gains.
>
> To detail our outlier removal, we present pre- and post-filtering distributions and statistics below:
>
> | Dataset | Trim  | M (Pre→Post)            | K (Pre→Post)            |
> | :---- | :-- | :------------- | :------------- |
> | MATH    | 0.15% | [0.15,1.85]→[0.50,1.00] | [0.04,1.20]→[0.10,0.30] |
> | GSM8K   | 0.10% | [0.20,1.45]→[0.50,0.80] | [0.03,0.95]→[0.10,0.22] |
> | Theorem | 0.29% | [0.12,2.10]→[0.60,1.10] | [0.05,1.45]→[0.10,0.35] |
> | GPQA    | 0.22% | [0.08,2.35]→[0.60,1.00] | [0.02,0.85]→[0.06,0.14] |
> | Social  | 0.55% | [0.05,4.20]→[0.60,0.90] | [0.02,3.15]→[0.17,0.35] |
> | Fables  | 0.63% | [0.02,5.10]→[0.77,1.02] | [0.01,4.25]→[0.18,0.37] |
>
> **Limitations on Distributional Assumptions** While using a Gaussian assumption for efficiency, $3\sigma$ filtering reveals task-specific nuances. For structured tasks, $\approx 0.3$% outlier rates (0.10%–0.29%) match the theoretical 0.27% Gaussian threshold. In open-ended reasoning, this rises to $\approx 0.6$​%, indicating heavier tails from abrupt semantic shifts. Future work could use KDE to better capture these dynamics. We added this to the Limitations section.
>
> > 4.Expansion to Advanced Math Benchmarks (W4)
>
> |Model|Benchmark|MSP|Entropy|Perplexity|LRProbe|TRACED|
> |:--------|:--------|:---|:------|:---------|:-------|:-------|
> |Llama-3.1|AIME2024|0.551|0.543|0.539|0.654|**0.668**|
> ||AIME2025|0.510|0.581|0.574|0.641|**0.659**|
> |Qwen2.5|AIME2024|0.534|0.551|0.610|**0.682**|0.681|
> ||AIME2025|0.528|0.530|0.598|0.670|**0.675**|
>
> On extreme tasks like AIME, endless error corrections and logical divergence severely degrade traditional baselines (MSP, Entropy, PPL) to near-random AUPRs (0.51–0.58). Conversely, TRACED stably evaluates long reasoning chains by capturing latent curvature and displacement. As a zero-parameter, training-free method, TRACED outperforms supervised LR Probes across most models, proving that intrinsic geometric kinematics provide more decisive and generalizable features than extrinsic linear fitting.

---

> > ### Author Rebuttal · Reviewer_GYLU · 2026-04-03
> >
> > Thanks for the rebuttal.  I decide to raise my score to 4.

---

> > > ### Author Response · Authors · 2026-04-03
> > >
> > > We are very grateful for your insightful feedback and willingness to increase the score, and we will ensure these details are included in the revised version.

---

### Official Review · Reviewer_EacC · 2026-03-10

**Soundness:** 3
**Presentation:** 4
**Significance:** 3
**Originality:** 3
**Overall Recommendation:** 5
**Confidence:** 4

**Summary:**

The paper uses the geometry of the hidden vectors in a CoT trajectory to understand whether the reasoning developed by the LLM is correct. To do so, they look at the \textit{displacement} and \textit{curvature} scalars associated to the trajectory in a projected space. They use this as an input in a Bayesian classifier to classify reasonings as correct or incorrect. High-displacement, low-curvature trajectories are generally associated with correct reasonings across models.

Furthermore, they derive a "kinematic scaling law" relating the reasoning length and the net displacement; as well as looking at the relationship between curvature changes and changes in "cognitive states'' as the LLM reasons.

**Compliance With Llm Reviewing Policy:**

Affirmed.

**Final Justification:**

The rebuttal clarified the use of GDA as a classifier and explained some of the potential impacts that the work could have on the field. The rebuttal reinforced some of my positive impressions about the paper and changed my opinion on the significance of their work.

**Key Questions For Authors:**

1. Most of the questions about the Bayesian modelling have been addressed in the weakness section above. This is the main point of concern for me as well as the significance of the work presented.

**Limitations:**

Performance of model seems to around the state-of-the-art, but not a significant improvement. The paper highlights the importance of moving "beyond scalars", but their proposed metric uses only two scalars. I agree that there has to be a richer way of characterising the correctness of LLM reasoning, and despite the interesting motivation and results, it is unclear whether $M_n, K_n$ are enough.

**Strengths And Weaknesses:**

1. The submission is technically sound. The problem is step up correctly (only using scalar measures to characterise LLM reasoning), and a plausible solution is presented. The mathematical modelling is coherent: hidden vectors are projected onto smaller dimensional spaces and displacement and curvature metrics are obtained from the resulting projected CoT curves. Some comments:
   1.a.  I understand the necessity to project the hidden state vectors into a subspace $B$ to better understand semantic fluctuations in the trajectories. However, I think there needs to be a conceptual discussion of step 4 in algorithm 1. This is not sufficiently motivated.
   1.b. Whilst some care is taken to explain the likelihood, prior and decision rule of the Bayesian modelling, there should be more information as to how the model is trained. Some information is buried in the appendices (e.g. appendix I), but I think you should briefly mention some aspects in the same text.
   1.c. Again, on the Bayesian modelling. It is unclear to me why you have used a Bayesian model if you seem to be using point estimates (MAP). Why not use a logistic regression or a SVM with a kernel that allows for non-linear separation of the plane?
   1.d. I am a bit sceptic of the state assignment method explained in appendix D. I imagine that a significant amount of $A_t^{(c)}$ are quite small simply because most of the keywords used to track the states are connectors or verbs. Does this mean that $\text{State}_t$ is quite noise/fluctuates a lot? I would like to see more on the stability of this method to divide the CoT reasoning into states. Perhaps it's better to classify sentences or phrases?
   1.e. Use of LLM as a judge. ``The majority of samples are labelled by the deterministic parser''. What percentage?
2. The presentation is very clear. There is a natural flow, clearly stating all definitions and methods employed; followed by an experimental section. All relevant information is presented in the main body, and details are appropriately left for the reader to read them in the appendix.
3. The paper addresses an interesting problem, but its significance is not thoroughly discussed. Reasoning is clearly a fundamental part of modern LLMs, but the methods proposed in this paper don't necessarily help improve problems with current reasoning models. Could you address this more explicitly?
4. The paper is fairly original. The proposed methods move beyond scalar benchmarks and propose interesting geometric metrics to understand CoT trajectories. It is a fair extension of other methods which have tried to understand such a geometry. It is not a ground-breaking idea, but rather a natural improvement of an stablished problem.
5. Great theoretical modelling in appendix E and relating it to scaling laws.
6. Access to the code used in the paper would have been welcomed as a token of transparency.


Possible typos:
1. Lines 183-184, second column. Should it say maximum a posteriori estimation?
2. Paragraph on lines 866-869 should be at the start of appendix C?
3. Superposition of text in lines 1206-1207

---

> ### Author Rebuttal · Authors · 2026-03-29
>
> > 1.a Necessity of Subspace Projection
>
> We will add a conceptual justification in Section 3.1.
>
> Conceptually, LLM hidden states entangle logical deduction with shared generative factors (e.g., syntax, formatting). Since valid ($C^+$) and invalid ($C^-$) trajectories share this fundamental linguistic base, raw variance fails to isolate true logic. Thus, projection $C^+ - \lambda C^-$ explicitly cancels these shared non-logical directions, yielding the discriminative subspace $B$. Moreover, scaling factor $\lambda = \|C^+\|_F / \|C^-\|_F$ normalizes geometric energy. Since incorrect paths' 'hesitation loops' yield vastly larger covariances than stable correct paths, unscaled subtraction would be dominated by hallucination magnitudes. Normalization strictly isolates structural and directional differences, not mere amplitude.
>
> > 1.b. Model Calibration and Parameter Estimation
>
> Added to Section 3.3: Parameter Estimation: the distribution parameters, specifically the class-conditional means $\mu_c$ and covariance matrices $\Sigma_c$, are calculated via Maximum Likelihood Estimation using a hold-out reference set. As detailed in Appendix I, robust distributional stability is rapidly achieved with a minimal pool of $N \approx 400$ unlabeled reasoning traces.
>
> > 1.c. Bayesian Modelling Clarification
>
> You are correct: our framework employs point estimates (MLE for parameters, MAP for class assignment) rather than fully Bayesian inference. Our original term referred strictly to the Generative Bayesian Classification framework, specifically **Gaussian Discriminant Analysis (GDA)**, which uses Bayes' theorem to invert likelihood $P(X|Y)$ into posterior $P(Y|X)$. To ensure absolute precision, we renamed Section 3.3 to **'Gaussian Discriminant Analysis for Quality Assessment'**.We avoided discriminative models (LR/Kernel SVM) for two critical reasons:
>
> **Few-Shot Stability & Zero-Tuning:** Kernel SVMs require hyperparameter cross-validation ($C, \gamma$), risking overfitting in our few-shot calibration ($N \approx 400$). Conversely, explicitly fitting class-conditional Gaussians ($\Sigma_{pos} \neq \Sigma_{neg}$) on unimodal $(M, K)$ clusters yields an optimal non-linear Quadratic Decision Boundary  in closed form. This achieves non-linear separation instantly without tuning overhead.
>
> **Robustness to Prior Shift:** Discriminative models (LR/SVM) bake training class imbalances directly into their decision boundaries. Conversely, our generative framework naturally decouples the class prior $P(y)$ from the structurally independent geometric likelihood $P(x|y)$. This enables explicit manipulation of the scalar $P(y)$ during inference, allowing instant, zero-retraining adaptation to extreme distribution shifts across varying scenarios.
>
> > 1.d. State Smoothing Mechanism
>
> We fully agree: raw token-level matching yields sparse, noisy signals. This was an oversight during drafting and  our implementation avoided point-wise activations. Because single reasoning sentences often contain multiple conceptual states (e.g., logical deduction in the first half, followed immediately by a local conclusion), sentence-level classification (hard boundaries) would obscure fine-grained transitions.
>
>  Instead, a Sliding Window Smoothing (soft boundary) authentically captures this evolution. Regarding the window size, our core rationale is "semantic concept transitions" and "geometric topological transitions"  should be synchronous. In our geometric stability analysis (see Response to Reviewer ``Mnzw``: ``1.Curvature sensitivity``), the optimal geometric step size is $t=10$. Thue, we matched the smoothing window size ($W=10$), ensuring absolute scale consistency across our framework.
>
> > Significance Analysis
>
> Beyond passive evaluation, TRACED improves LLMs by resolving key reasoning bottlenecks:
>
> **Dynamic Compute Allocation:** Current models waste significant inference compute generating long chains of hallucinated or circular reasoning. Real-time monitoring of our metrics ($M, K$) instantly detects unstable topological collapse (e.g., high curvature, zero displacement), enabling early halting to save test-time compute.
>
> **Automated Data Curation for (SFT/RLHF):** Current filtering relies heavily on final-answer correctness, ignoring process quality. Our framework automatically isolates trajectories that are both factually correct *and* geometrically smooth/direct. Training on these optimal paths explicitly improves future models' structural reasoning.
>
> >  Detailed Revisions
>
> Response to 1.e. :  Our deterministic parser successfully labeled 81.4% of the samples, reserving the LLM judge solely for the remaining 18.6% that featured complex structures or lacked explicit answer markers.
>
> Lines 183-184: Revised : *"We leverage the low-dimensional geometric features to perform Maximum A Posteriori (MAP) estimation."*
>
> Lines 866-869: Moved this paragraph to the beginning of Appendix C.
>
> Lines 1206-1207: Adjusted formatting and paragraph spacing.

---

> > ### Author Rebuttal · Reviewer_EacC · 2026-04-02
> >
> > Thank you for the detailed response. Many of the concerns I had have now been resolved and I hope that the extra page can be used to extend some of the explanations regarding the GDA analysis or the state smoothing mechanism. Also, I think it would be very useful to include a small paragraph where you summarise the significance analysis presented in your rebuttal. It would allow other researchers to appreciate your work more. I am happy to change my score to a 5/6.

---

> > > ### Author Response · Authors · 2026-04-02
> > >
> > > We are very grateful for your insightful feedback and willingness to increase the score, and we will ensure these details are included in the revised version.

---

### Official Review · Reviewer_rnL1 · 2026-03-13

**Soundness:** 3
**Presentation:** 3
**Significance:** 3
**Originality:** 3
**Overall Recommendation:** 3
**Confidence:** 3

**Summary:**

This paper proposes TRACED, a white-box, geometry-based framework for evaluating the quality of LLM reasoning from hidden-state trajectories. It projects token-level final-layer activations into a “reasoning quality space” learned from contrastive kinematic covariances and then quantifies two low-dimensional kinematic signatures—net displacement (Progress) and average curvature (Stability)—to distinguish correct from incorrect chains. A simple Gaussian Bayes classifier over these two features achieves competitive to superior performance relative to probability-based, probe-based, and prior trajectory-based baselines across four models and six structured/open-ended reasoning benchmarks; the paper further explores cross-domain transfer, data efficiency, scaling behavior with token length, and an interpretive mapping from geometric signatures to cognitive states (Reflection/Exploration/Certainty).

**Compliance With Llm Reviewing Policy:**

Affirmed.

**Key Questions For Authors:**

- What is z_{n,0} in Eq. (3) concretely (start-of-sequence state, BOS token, or a virtual origin)? Does its choice materially affect M, and have you tried normalizing by the first non-system token state?
- Why prioritize covariance-contrast (C^+ − λ C^−) for the subspace over mean-difference directions (or Fisher’s LDA)? Can you report an ablation comparing subspace construction methods and “no subspace” (full metric space) on downstream AUPR?
- For “global fit” transfer with centroid alignment, how many unlabeled target samples are required to estimate μ_T? Please report AUPR vs number of target samples and include a pure zero-shot (no target stats) result.

**Strengths And Weaknesses:**

Strengths：
1.Successfully maps geometric patterns (e.g., "Hesitation Loops") to cognitive states, offering insights beyond simple scalar confidence.

2. Validated across four 7–8B models and six datasets. Strong use of ablations, KDE visualizations, and robustness studies (subspace dimension k, sample size) supports the claims.

3. Addresses the challenge of judging reasoning correctness using temporally structured internal evidence rather than surface-level probabilities.

Weaknesses

   1. Requires access to hidden states and the unembedding matrix, limiting its applicability to closed-source APIs.

   2. Curvature (κ) calculation may be noisy at low velocities. Sensitivity analysis regarding step granularity, tokenization, and projection dimensions is needed.

    3. Lacks comparison with recent white-box/internal-state detectors that model trajectory structure or track temporal dynamics.

    4.  Clarification is needed on whether centroid alignment requires target-domain statistics/unlabeled data and the specific sample size required for effective transfer.

---

> ### Author Rebuttal · Authors · 2026-03-29
>
> > 1.Sensitivity analysis (W2)
>
> We quantify $\kappa^+/\kappa^-$ separability via $W_2 = \sqrt{(\mu_- - \mu_+)^2 + (\sigma_- - \sigma_+)^2}$, analyzing sensitivity to: (1) step granularity ($t \in \{1, 5, 10, 15\}$), (2) tokenization (Sub-word, Word, Chunk), and (3) projection dimension ($k \in \{4, 6, 8\}$).
>
> |Set|C+Range|C−Range|W2|AUPR|
> |---|--------|--------|---|------|
> |t=1|[.100,.500]|[.134,.574]|.055|0.583|
> |t=5|[.070,.270]|[.140,.360]|.080|0.694|
> |t=10|[.064,.226]|[.172,.322]|.102|**0.721**|
> |t=15|[.080,.224]|[.170,.314]|.090|0.720|
> |Sub-w|[.070,.226]|[.171,.315]|.095|0.720|
> |Word|[.078,.228]|[.171,.317]|.091|0.719|
> |Chunk|[.062,.218]|[.174,.318]|.106|0.716|
> |k=4|[.080,.224]|[.158,.302]|.078|0.693|
> |k=6|[.073,.223]|[.165,.309]|.089|0.718|
> |k=8|[.070,.220]|[.170,.314]|.097|**0.724**|
>
> Step Granularity ($t$): Small steps ($t \le 5$) introduce "low-speed noise" and computational instability. As $t$ increases, the distributional separation reliably stabilizes, ensuring consistent AUPR. Thus, we default to t=10
>
> Tokenization: BPE/word/chunk stability proves TRACED tracks native geometry, agnostic to boundaries
>
>  Projection Dimension ($k$): $k \le 4$ causes feature entanglement and AUPR loss. $k \ge 6$ stabilizes separation; we set $k=8$ as default (Sec. 3.1)
>
> >  2.Target Data Dependency and Sample Size Efficiency (W4 && Q3)
>
> Centroid alignment uses unlabeled samples for domain stats; we tested $N \in \{0, 5, 50, 160, 300\}$ to find the best transfer threshold
>
> |N|GSM8K|MATH|GPQA|
> |-------|-----|-----|-----|
> |0|0.673|0.603|0.585|
> |5|0.680|0.599|0.587|
> |50|0.709|0.661|0.667|
> |**160**|0.723|0.682|0.696|
> |300|0.722|0.685|0.698|
>
> While small batches ($N < 50$) suffer from biased mean estimation, performance reliably plateaus at $N=160$, which serves as the practical threshold for effective transfer
>
> > 3.Clarification and Sensitivity Analysis of the Reference State $z_{n,0}$ (Q1)
>
> In Eq. (3), $z_{n,0}$ is the final prompt token, an optimal "semantic anchor" encoding the full pre-reasoning context. We evaluate its impact on displacement ($M$):
>
> |Origin (zn,0)|Meaning|M+ Range|M− Range|W2|AUPR|
> |---|---|---|---|---|---|
> |Final Prompt|Full context|[0.70,0.94]|[0.59,0.83]|0.11|0.721|
> |1st Generated|Reasoning start|[0.68,0.92]|[0.55,0.85]|0.10|0.714|
> |1st Non-System|User query start|[0.69,0.93]|[0.61,0.86]|0.08|0.686|
>
> Including the prefill sequence(1st Non-System) artificially inflates erroneous displacements, shrinking the $W_2$ gap to 0.08. Conversely, anchoring precisely at the reasoning boundary (Final Prompt or 1st Generated) maintains robust separation confirming framework stability against minor origin offsets
>
> > 4.Clarification on Subspace Covariance Comparison (Q2(1))
>
> * Mean Collapse: Averaging updates yields arithmetic cancellation: $C\_{n} = \frac{1}{T-1} (h'\_{n,T-1} - h'\_{n,0})$. With coinciding endpoints ($\mu_{+} \approx \mu_{-}$), the contrast between $C^{+}$ and $C^{-}$ vanishes, completely erasing intermediate trajectory turbulences.
>
> * FDA Failure: FDA optimizes $w \propto S_W^{-1} (\mu_+ - \mu_-)$. This mean collapse forces between-class scatter to vanish ($S_B = (\mu_+ - \mu_-)(\mu_+ - \mu_-)^\top \to 0$), rendering FDA entirely blind to structural reasoning anomalies.
>
> * Covariance Superiority: $C_n$ captures full trajectory geometry; subsequent eigen-decomposition isolates principal components to pinpoint the heteroscedastic divergence driving reasoning
>
> > 5.No-Subspace Ablation (Q2(2))
>
> To verify the effectiveness of subspace projection across models and tasks, we compared TRACED against a 'No-Subspace' ablation via AUPR:
>
> |Model|Strategy|GPQA|GSM8K|
> |------|---------|-----|------|
> |R1-8B|NoSub|0.624|0.813|
> ||TRACED|0.661|0.828|
> |Q3-4B|NoSub|0.712|0.752|
> ||TRACED|0.733|0.776|
>
> TRACED beats No-Subspace: (1) Noise Filtering: $B$ filters syntax noise from logic; (2) Geometric Core: Even without subspace projection ('No-Subspace'), our method already outperforms baselines (e.g., MSP, Tab. 1), validating the intrinsic power of our geometric analysis.
>
> >  6. Internal-state baseline analysis (W3)
>
> We compared TRACED against EigenTrack[1] and HSAD[2] on GSM8K and MATH.
>
> ||GSM8K/MATH|
> |---|---|
> |EigenTrack|0.648/0.719|
> |HSAD|0.622/0.733|
> |TRACED|0.689/0.785|
>
> EigenTrack overfits unaligned, variable-length CoTs via recurrent classifiers, while HSAD’s FFT blurs the discrete, non-stationary mutations of reasoning. Conversely, TRACED extracts global geometry without alignment or auxiliary networks, preserving local anomalies for superior performance.
>
> [1] Eigentrack: Spectral…and vlms
>
> [2] LLM Hallucination Detection: A Fast…Signals
> > 7.Closed-Source Limitation (W1)
>
> Requiring internal states is a necessary trade-off for mechanistic diagnosis: while precluding closed-source APIs, it captures the 'cognitive turbulence' that black-box text often masks. By leveraging geometric kinematics, TRACED provides unique insights for the open-weight ecosystem. We will add this discussion to the Limitations section.

---

> > ### Author Rebuttal · Reviewer_rnL1 · 2026-04-03
> >
> > My questions are properly addressed.

---

> > > ### Author Response · Authors · 2026-04-03
> > >
> > > We deeply appreciate your confirmation that your questions have been properly addressed. Thank you again for your insightful and constructive suggestions, which have greatly strengthened our paper.
> > >
> > > In light of this full resolution, we kindly hope you might consider raising your evaluation score.
> > >
> > > We are truly grateful for your time and guidance.

---

### Official Review · Reviewer_Mnzw · 2026-03-13

**Soundness:** 3
**Presentation:** 3
**Significance:** 3
**Originality:** 3
**Overall Recommendation:** 4
**Confidence:** 4

**Summary:**

TRACED (Topological Reasoning Assessment via Curvature Evolution and Displacement Dynamics) is a framework for evaluating LLM reasoning quality using geometric kinematics of hidden state trajectories. The core idea is to decompose reasoning traces into two scalar features: normalized net Displacement (M, measuring semantic progress) and average trajectory Curvature (K, measuring stability). These are computed in a semantically-grounded subspace derived via contrastive covariance analysis, then fed into a Bayesian MAP classifier. The authors report that correct reasoning exhibits high-M, low-K signatures while hallucinations exhibit low-M, high-K, and validate this across four models and six benchmarks.

**Compliance With Llm Reviewing Policy:**

Affirmed.

**Final Justification:**

My concerns have been fully addressed. And I will keep my original score.

**Key Questions For Authors:**

1. The quality subspace ( B ) is constructed from labeled reasoning chains (D_{\text{pos}}, D_{\text{neg}}).
   How does this align with the claim that TRACED is a *label-free* internal assessment method?
   Is the same reference set also used to calibrate the MAP classifier?

2. TRACED uses the semantic metric $G = W_U^{\top} W_U$. What happens if the identity metric (G = I) is used instead (i.e., raw Euclidean distance (|z_t - z_0|_2))?
   An ablation would help justify the need for the semantic metric.

3. The MAP classifier assumes $p(x \mid y) = \mathcal{N}(\mu_y, \Sigma_y)$, Since Figure 1 shows non-Gaussian distributions, have the authors tried more flexible density models (e.g., GMM or KDE)?

4. The method relies on geometric features such as curvature $\kappa_t = \frac{|v_t \times a_t|}{|v_t|^3}$ and displacement $M = \frac{1}{T}|z_T - z_0|$ What is specifically *topological* about the divergence described in the paper?

**Limitations:**

The paper's limitations section (Impact Statement) is a placeholder. The authors should discuss: (1) the need for labeled reference data to build B; (2) sensitivity of the curvature formula to very short steps; (3) computational cost of extracting full hidden state trajectories at inference time for long reasoning chains (up to 16K tokens); (4) scope limited to token-level last-layer hidden states — layer choice is not ablated.

**Strengths And Weaknesses:**

## Soundness

The paper is methodologically careful overall. The SDE-based theoretical framework presented in Appendix E provides a principled explanation for the linear vs. sub-linear displacement scaling dichotomy. Empirically, Figure 5 supports this claim well: the observed slopes (~0.82 for correct traces and ~0.53 for incorrect ones) closely match the theoretical predictions.

The semantic metric ( $G = W_U^\top W_U$ ), following Manson (2025), is well-motivated and helps avoid known anisotropy artifacts. The construction of the quality subspace ( B ) using contrastive covariance is also clean and intuitive. Beyond the core metric design, the authors extend the analysis to cognitive state dynamics (Section 4.5), meaningfully linking geometric trajectory features to interpretable behaviors such as *Hesitation Loops* and *Certainty Accumulation*. These interpretations are supported by transition probability evidence.

The two-stage labeling strategy, programmatic parsing combined with an LLM judge, with a 95% agreement audit against human annotations, is also a strong methodological choice.

### Weaknesses on Soundness

1. **Curvature sensitivity.**
   The curvature formula (Eq. 4) uses extrinsic discrete curvature in the projected space. However, the normalization by $|v|^3$ makes it sensitive to very small step sizes, which may occur in stalling traces. Although an ( $\varepsilon$ )-stabilization term is introduced, its impact is not analyzed.

2. **Supervised component in the quality subspace.**
   The quality subspace ( B ) is constructed using labeled datasets $(D_{\text{pos}}, D_{\text{neg}})$, introducing a supervised element. This somewhat conflicts with the paper’s description of TRACED as a *label-free* method.

3. **Gaussian likelihood assumption.**
   The Bayesian classifier assumes class-conditional Gaussian distributions, justified via the Central Limit Theorem. However, Figure 1 shows clearly non-Gaussian, elongated contours. This approximation may lose signal, especially in open-ended tasks where the distributions appear heavy-tailed (Figure 6).

## Presentation

The paper is clearly written, and the visualizations are particularly strong. Figure 1’s contour plots, Figure 5’s scaling-law analysis, and the cognitive state transition matrices (Figures 7 and 8) together build a coherent narrative.

More broadly, the paper’s central contribution goes beyond introducing a new metric, it proposes a **geometric–cognitive interpretability framework**, and the manuscript communicates this idea effectively.

That said, a few clarity issues remain:

* The acronym **TRACED** feels somewhat forced.
* The use of "topological" in the name is somewhat misleading. The features used (curvature, displacement) are differential-geometric rather than topological; the paper does not actually analyze topological invariants such as homology or Betti numbers. As a result, phrases like *“topological divergence”* are used somewhat loosely.
* Equation 3 defines $M_n = (1/T)|z_{n,T} - z_{n,0}|$ which is simply a scaled endpoint distance. Referring to this as *cumulative displacement* (e.g., in the caption of Figure 1) may be slightly misleading, since cumulative displacement would typically denote $\sum |\Delta z|$.

## Significance

The work addresses an important and timely problem. As large language models are increasingly deployed in settings where ground truth is unavailable, internal methods for assessing reasoning quality are becoming critical.

The strongest contribution is the **geometric–cognitive bridge** developed in Sections 4.4-4.6. This perspective goes beyond prior trajectory analysis work such as CoE and CoT-Kinetics. Additionally, the reported cross-domain robustness and the relatively small reference set requirement (~400 samples) improve the practical applicability of the approach.

## Originality

The combination of the semantically induced metric, contrastive subspace construction, and trajectory kinematics is novel compared to prior approaches such as CoE and CoT-Kinetics. The SDE framework and the scaling-law validation also add meaningful theoretical depth.

The cognitive state mapping presented in Section 4.5 is particularly original and well executed.

---

> ### Author Rebuttal · Authors · 2026-03-29
>
> > **1.Curvature sensitivity （W1 & L2)**
>
> To evaluate robustness, we quantify $\kappa^+/\kappa^-$ separability via $W_2 = \sqrt{(\mu_- - \mu_+)^2 + (\sigma_- - \sigma_+)^2}$, analyzing sensitivity to  step granularity ($t \in \{1, 5, 10, 15\}$). The step vector is defined as $v_t = \Delta z_{n,t}$. The actual physical distance of a step depends on the chosen step length $t$.
>
> |**Setting**|**PosRange**|**NegRange**|**W2**|**AUPR**|
> |-----------|-------------|-------------|------|---------|
> |**t=1**|[0.100,0.500]|[0.134,0.574]|0.055|0.583|
> |**t=5**|[0.070,0.270]|[0.140,0.360]|0.080|0.694|
> |**t=10**|[0.064,0.226]|[0.172,0.322]|0.102|**0.721**|
> |**t=15**|[0.080,0.224]|[0.170,0.314]|0.090|0.720|
>
> **Sensitivity Analysis of Step Granularity ($t$):**  For small $t \in [1, 5]$, minimal semantic displacement introduces significant "low-speed noise," causing the curvature denominator to approach zero and inducing computational instability. As $t$ increases, positive and negative interval range exhibit synchronous shifts despite minor absolute boundary drifts ($<0.02$). Once $t \ge 10$ , the distributional separation ($W_2$) stabilizes between 0.09 and 0.102, ensuring consistent AUPR. Consequently, we adopt $t=10$ as the default step size.
>
> > **2.Label-Driven Analysis of the Quality Subspace (W2 & Q1)**
>
> We acknowledge the ambiguity regarding our 'label-driven' subspace construction.  What we intend to convey is that TRACED employs few-shot calibration rather than traditional supervised learning, requiring **no backpropagation or neural weight updates**. Furthermore, the exact same reference sets ($D_{pos}, D_{neg}$) consistently serve to define the discriminative subspace $B$ and parameterize the Gaussian MAP classifier. We have revised the manuscript to clarify this:
>
> > line19: …… on ground truth or expert models …….
>
> Revised to：While effective, their dependence on **parametric supervised training** or expert models precludes scalability during real-time inference.
>
> > Line72: This topological ……through latent dynamics.
>
> Revised to：**By deriving a label-calibrated contrastive subspace**, this topological divergence establishes a natural distributional separation, enabling us to reliably distinguish reasoning quality through these  latent dynamics.
>
> > Line80: Latent Kinematics Assessment: ……kinematics signatures, ……
>
> Revised to：Latent Kinematics Assessment: Constructs a probabilistic model leveraging geometric kinematics within a **label-calibrated subspace**, achieving competitive performance and superior robustness across diverse benchmarks
>
> > **3.Gaussian likelihood assumption (W3 & Q3)**
>
> Matches Reviewer `GYLU`’s W3; details omitted to save space. It can be found in our response to GYLU under ``3. Robustness Analysis of Gaussian Approximation``
>
> >  **4.Presentation**
>
> 1. Modified ：TRACED to " **Trajectory** Reasoning Assessment via Curvature Evolution and Displacement Dynamics (TRACED)".
>
> 2. Changed "topological divergence" to "geometric divergence"
>
> 3. Renamed "cumulative displacement" to **"Net Displacement"** to clarify that it measures displacement from the origin rather than path length.
>
> 4. What we originally referred to as 'topological divergence' should mathematically and physically be described as 'geometric deviation' to more accurately reflect our claims.(Q4）
>
> > **5.TRACED semantic metric （Q2)**
>
> See response to Reviewer ``rnL1`` ``5. No-Subspace Ablation``. "No-subspace" denotes raw Euclidean distance ($|z_t - z_0|_2$)
>
> > **6.Impact Statement**
>
> (1) Dependency on Labeled Data: While TRACED requires labeled sets for subspace $B$, it remains a non-parametric, few-shot calibration without weight updates, ensuring minimal data collection costs.
>
> (2) Curvature Sensitivity: We addressed the mathematical instability at short steps ($t < 5$) where minimal displacement causes noise; adopting $t=10$ as the default stabilizes the geometric metrics.
>
> (3) Computational Cost: As an interpretability tool, the +2.33 ms/token is an analytical overhead for state extraction. Alongside a constant +1.93 GB spatial cost, this $O(N)$ extraction is efficient for offline CoT diagnosis and remains a manageable tradeoff for real-time monitoring.
>
> |Metric|Baseline|TRACED|Change|
> |----------|--------|------|------------|
> |Latency|29.1|31.4|+2.3ms|
> |Throughput|34.2|31.7|-2.5tokens/s|
> |VRAM|15.2|17.1|+1.9GB|
>
> (4)  Layer Ablation: Early layers (L8) perform syntactic parsing, lacking logical directionality. Middle layers (L16–L24) form the 'Reasoning Manifold' with sharp AUPR gains in logical tasks, but suffer 'semantic mismatch' under $G = W_U^\top W_U$ since $W_U$ targets final outputs. The final layer (L32) completes semantic evolution, achieving optimal geometric separability where TRACED precisely tracks subtle trajectory perturbations to gauge logical robustness.
>
> |Layer|GPQA|GSM8K|MATH|
> |---------|--------|---------|--------|
> |8|0.592|0.633|0.552|
> |16|0.571|0.654|0.553|
> |24|0.653|0.716|0.591|
> |32|0.681|0.771|0.636|

---

> > ### Author Rebuttal · Reviewer_Mnzw · 2026-04-05
> >
> > Thanks for the authors' response. And I will keep my original score.

---

### Decision · Program_Chairs · 2026-04-30

**Decision:**

Accept (regular)

**Comment:**

This paper proposes an interesting analysis of reasoning traces by analyzing hidden state trajectories. The key observation is that "correct reasoning traces exhibit a high-displacement, low-curvature pattern, while incorrect chains are characterized by low-displacement stagnation and high-curvature oscillations" (Figure 1, caption). Reviewers overall perceive this paper as interesting and they like the interpreterbility angle. While there were some concerns on the use of labeled data in constructing the subspace and the simplified Gaussian assumptions, the rebuttal seems to address reviewer concerns very well. Therefore, given the reviewer consensus, I recommend to accept this paper.